# Circ-CREBBP inhibits sperm apoptosis via the PI3K-Akt signaling pathway by sponging miR-10384 and miR-143-3p

Ning Ding [1,4], Yu Zhang[1,4], Mengna Huang[1], Jianfeng Liu[1], Chonglong Wang[2], Chun Zhang[1], Jinkang Cao[1], Qin Zhang[1,3] & Li Jiang [1✉]

Male reproductive diseases are becoming increasingly prominent, and sperm quality is an important indicator to reflect these diseases. Seminal plasma extracellular vesicles (SPEVs) are involved in sperm motility. However, their effects on sperm remain unclear. Here, we identified 222 differentially expressed circRNAs in SPEVs between boars with high or low sperm motility. We found that circ-CREBBP promoted sperm motility and inhibited sperm apoptosis by sponging miR-10384 and miR-143-3p. In addition, miR-10384 and miR-143-3p can regulate the expression of *MCL1*, *CREB1* and *CREBBP*. Furthermore, we demonstrated that MCL1 interacted directly with BAX and that CREBBP interacted with CREB1 in sperm. We showed that inhibition of circ-CREBBP can reduce the expression of MCL1, CREB1 and CREBBP and increase the expression of BAX and CASP3, thus promoting sperm apoptosis. Our results suggest that circ-CREBBP may be a promising biomarker and therapeutic target for male reproductive diseases.

[1] National Engineering Laboratory for Animal Breeding, Key Laboratory of Animal Genetics, Breeding & Reproduction, Ministry of Agriculture, College of Animal Science & Technology, China Agricultural University, 100193 Beijing, P. R. China. [2] Key Laboratory of Pig Molecular Quantitative Genetics of Anhui Academy of Agricultural Sciences, Anhui Provincial Key Laboratory of Livestock and Poultry Product Safety Engineering, Institute of Animal Husbandry and Veterinary Medicine, Anhui Academy of Agricultural Sciences, 230031 Hefei, P. R. China. [3] College of Animal Science and Technology, Shandong Agricultural University, 271018 Tai'an, P. R. China. [4]These authors contributed equally: Ning Ding, Yu Zhang. ✉email: lijiang@cau.edu.cn

Circular RNAs (circRNAs), a class of noncoding regulatory RNAs, are characterized by a special mode of splice junction, in which the 5′- and 3′-ends of a single exon or of multiple adjacent exons are joined and released in a circular form[1,2]. CircRNAs are unusually stable due to circular configuration and ability of resistance to exoribonucleolytic degradation[3]. In addition, circRNAs are specifically expressed in cells of different kinds[1] and generally localized in the cytoplasm or stored in extracellular vesicles (EVs)[4,5]. Recent studies have shown that EVs contain more circRNAs than secretory cells[3,4]. These characterizations may enable circRNAs to have promising diagnostic and therapeutic value in diseases[6,7]. CircRNAs were found to be markers for the molecular diagnosis of many cancers, such as bladder cancer[8], breast cancer[9] and hepatocellular carcinoma[10].

Many studies have verified the interplay between circRNAs and reproductive stem cells, sperm, embryonic development and reproductive diseases. For instance, Li et al. found 18,822 circular RNAs in mouse germ cells, including 245 male-specific circRNAs and 676 female-specific circRNAs[11]. Dong et al. explored 15,996 circRNAs from human testis, of which 67% of the circRNA host genes are closely related to spermatogenesis, sperm motility and fertilization[12]. However, these circRNAs were not only specifically expressed in the testis but also detected in seminal plasma with high stability. In addition, many studies have shown that abnormal expression of circRNAs plays significant roles in non-invasive diagnosis and therapeutic intervention for reproductive diseases and prostate cancer (PCa)[13]. A recent study showed that hsa_circRNA_0023313 was obviously upregulated in testicular tissue of nonobstructive azoospermia (NOA) patients[14]. In addition, circ-SLC19A1 was increased in PCa cells and their secreted EVs[15].

Extracellular vesicles are present in most fluids, including milk, urine, tears, serum and plasma. EVs contain many bioactive molecules, such as RNA, DNA, proteins and lipids[16]. Many studies have shown that EVs can transfer their cargo to recipient cells, which playing an important role in the process of intercellular communication[17]. Increasing evidence has shown that EVs can regulate signal transduction pathways and act as mediators of signal crosstalk[18]. Seminal plasma contains a large number of EVs produced and released by the prostate epithelium, epididymal epithelial cells, and other accessory glands[19,20]. It has been demonstrated that seminal plasma extracellular vesicles (SPEVs) can bind sperm in vitro, promote sperm motility, prolong sperm survival, improve the integrity and antioxidant capacity of sperm plasma membrane[21]. Many studies have shown that RNA molecules carried by SPEVs play an important role in the process of sperm maturation, sperm motility and sperm-egg binding[22–25]. Thus, the function of SPEVs and their molecular content have become a new focus of current research.

Spermatogenesis and sperm survival are powerfully regulated by specific genetic expression[1,26]. We recently showed that miR-222 in SPEVs plays an important role in sperm survival[24]. It has also been reported that some miRNAs in SPEVs can be used as markers for the diagnosis of human azoospermia[27] or prostate cancer[25,28], such as miR-31-5p, miR-142-3p, miR-142-5p, miR-223-3p, miR-27a-3p, miR-27b-3p, miR-155-5p, and miR-378a-3p. As a major participant, circRNAs play a role of post-transcriptional regulators by sponging miRNAs and interacting with RNA binding proteins[1,29]. The role of circRNA as a miRNA sponge in regulating the expression of target genes has been verified in many studies. For example, Kong et al. found that circFOXO3 can upregulate *SLC25A15* by acting as a miR-29a-3p sponge, regulating the cell cycle and cell apoptosis in PCa[30]. Chen et al. found that circHIPK3 promoted the invasion of prostate cancer by sponging miR-193a-3p[31]. In addition, some studies

have found that circRNAs regulate tumor progression through multiple signaling pathways[15,32].

Although there are many studies on circRNAs, knowledge of the expression levels and functions of EV-derived circRNAs in mammalian reproductive systems remains very limited. To our knowledge, there is no comprehensive study on circRNAs of SPEVs in mammals. It has been reported that the amino acid homology between pigs and humans is 84.1%, and they are very similar in morphological structure and physiological function[33]. Therefore, pigs are considered to be an ideal animal model for the research of human diseases[34,35]. Here, we investigated the SPEVs of boars with high and low sperm motility using whole transcriptome sequencing and identified 222 differentially expressed circRNAs (DECs). Among these DECs, circ-CREBBP was one of the top 5 circRNAs with the largest differential expression between groups. Based on the circRNA-miRNA-mRNA network, we found that circ-CREBBP targets miR-10384 and miR-143-3p, and miR-10384 and miR-143-3p target *MCL1*, *CREB1* and *CREBBP*, which participate in the PI3K-Akt signaling pathway. Therefore, we studied the mechanism by which circ-CREBBP is involved in the regulation of sperm motility. The results showed that circ-CREBBP could function as a ceRNA by harboring miR-10384 and miR-143-3p to abolish the suppressive effect on the target genes *MCL1*, *CREB1* and *CREBBP*, which promoted sperm survival and motility. Our study demonstrated the important role of circ-CREBBP in sperm motility, and the circ-CREBBP/miR-10384 and miR-143-3p/MCL1, CREB1 and CREBBP signaling pathways might be promising diagnostic biomarkers and therapeutic targets for male reproductive diseases.

## Results

**Characterization of boar SPEVs.** Phenotype analysis showed that there were significant differences in sperm motility between two groups. The average total sperm motility of H group was 0.94, while that of L group was 0.40 (Fig. 1a). SPEVs were isolated from the boars in two groups by the ultracentrifugation method. The results of transmission electron microscopy showed that SPEVs appeared intact and typically cup shaped (Fig. 1b and Supplementary Fig. 1). The mean size of the SPEVs was 124.1 nm, and the particle size ranged from 20 to 300 nm (Fig. 1c). The EV markers, Alix, Tsg101, CD9 and CD81 were all detected in SPEVs from four boars in two groups. In contrast, Calnexin, a negative marker of EVs, was not present in the SPEVs (Fig. 1d and Supplementary Fig. 1).

**CircRNA identification in SPEVs.** A total of 17,379 circRNAs were identified from all samples by using find_circ and CIRI2 software. Compared with the circNET (http://lnc.rnanet.org/circ/) and circAtlas databases (http://circatlas.biols.ac.cn/), we found that 80% of the circRNAs in our results were known circRNAs, and 3578 circRNAs were novel (Fig. 2a). Among the circRNAs, 84.9% were exonic circRNAs, 6.3% were intronic circRNAs, and 8.7% were intergenic region circRNAs. Exonic circRNAs accounted for a very large proportion, from 85.8 to 91.8% of all samples (Fig. 2b). The identified circRNAs were concentrated on chromosomes 1, 3, 6, and 13. The number of circRNAs on chromosome Y was the lowest. It was noted that 125 circRNAs were derived from mitochondria (Fig. 2c). The expression levels of circRNAs originating from exons, introns and intergenic regions did not show significant changes (Fig. 2d). A total of 15,860 circRNAs were generated from 4,996 paternal genes. Among them, 40.89% of paternal genes only produced one circRNA isoform, while other genes could produce multiple circRNA isoforms (Fig. 2e). The results showed that 24 genes produced 20 or more circRNA isoforms (Supplementary Table 1).

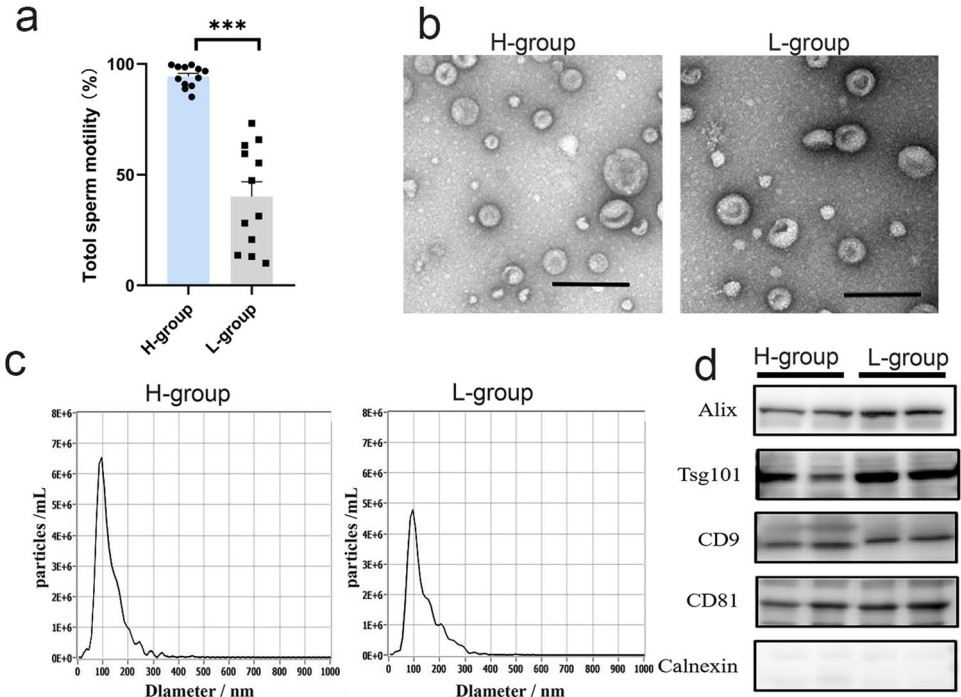

**Fig. 1 Isolation and characterization of SPEVs from boars. a** The difference in total sperm motility between the two groups. ***$P \leq 0.001$ (Student's $t$ test), $n = 12$ per group. **b** TEM image of SPEVs. Scale bars: 200 nm. **c** NTA results showing that the SPEVs were approximately 20–300 nm in diameter. **d** Western blotting results of SPEVs. Alix, Tsg101, CD9, and CD81, which are EV markers, were expressed in isolated SPEVs, and the negative marker Calnexin was not found in our isolated SPEV samples. H-group represents high sperm motility group and L-group represents low sperm motility group.

For example, the paternal gene *SMARCC1* could produce 44 circRNA isoforms. Interestingly, the expression of some circRNA isoforms produced by these host genes was very high. The top 30 most abundant circRNAs are listed in Table 1.

**Conservation and sequence characterization of circRNAs**. A total of 9520 detected circRNAs corresponding to human homologous circRNAs could be found in the circAtlas and circBank databases. Furthermore, 5045 circRNAs showed no nucleotide difference in back-splice junction (BSJ) contexts (±5 nucleotides). Among them, 2195 circRNAs had a higher conservation between pigs and humans, with a PhastScore >0.3 (Supplementary Data 1). In addition, there was no significant difference in the conservative analysis scores of different types of circRNAs (Fig. 2f). We observed many reverse complementary matches (RCMs) in the two introns flanking the BSJ sites of exonic circRNAs. Of note, some repeat sequences extensively existed in the RCM region. In particular, short interspersed nuclear elements (SINEs) and simple_repeat accounted for 87.72% and 8.73% of all identified RCMs, respectively (Fig. 2g and Supplementary Data 2).

**Identification of differentially expressed circRNAs**. A total of 222 DECs were detected between the H and L groups (Supplementary Data 3). Most of the DECs (185 out of 222) showed higher expression in boars in L group (Fig. 3a). Among these DECs, circ-ENSSSCG00000011850 (13:134233747|134234461) exhibited the highest upregulation ($\log_2$FC = 9.51) in L group, and circ-16:43997828|44055338 exhibited the lowest downregulation ($\log_2$FC = −9.16) in L group. The top 20 differentially expressed circRNAs are shown in Table 2. For miRNAs, ssc-miR-146a-5p exhibited the highest upregulation ($\log_2$FC = 4.62) in L group, and ssc-miR-206 exhibited the lowest downregulation ($\log_2$FC = −2.74) in L group (Fig. 3b). To further investigate the potential functions of the host genes of the identified DECs,

enrichment analysis was conducted by using KOBAS. The enrichment analysis revealed 25 significant GO terms (FDR $P < 0.05$) (Supplementary Table 2) and 19 significant pathways related to the DEC host genes (FDR $P < 0.1$) (Fig. 3c). Pathway analysis revealed that the PI3K-Akt signaling pathway, FoxO signaling pathway, autophagy, prostate cancer, and Wnt signaling pathway were the predominant biological processes represented.

**Experimental validation of SPEV-derived circRNAs**. To avoid false prediction of circRNAs and verify the sequencing results, six circRNAs (circ-CREBBP, circ-EP300, circ-KLHL3, circ-SUGCT, circ-PTGES3, and circ-SLC5A10) were randomly selected for further validation in H-SPEVs and L-SPEVs. We performed reverse transcription PCR with random primers but did not use the canonical linear sequence of the host gene and successfully detected the back splicing of circRNA in SPEVs. For each candidate circRNA, the expected sizes of gel-extracted bands were obtained, and their correct head-to-tail junction sequences were confirmed by Sanger sequencing (Fig. 3d). Furthermore, the results showed that circ-CREBBP, circ-EP300, circ-KLHL3, circ-SUGCT, and circ-SLC5A10 were significantly differentially expressed between the two groups (Fig. 3e), and the expression trends in the two groups were consistent with the sequencing results.

**Establishment of the circRNA-miRNA-mRNA network**. According to the results of DECs targeting miRNAs and the opposite expression trend of circRNAs and miRNAs, 106 interaction pairs of DECs-DEMs were obtained, including 37 circRNAs and 19 miRNAs (Fig. 4a, b and Supplementary Data 4). In particular, we constructed the regulatory networks of circ-CREBBP and circ-EP300, their targeted miRNAs, the target genes of miRNAs and the regulatory pathways (Fig. 4c). The results showed that circ-CREBBP simultaneously targets miR-10384 and miR-143-3p, and circ-EP300 targets miR-96-5p. In

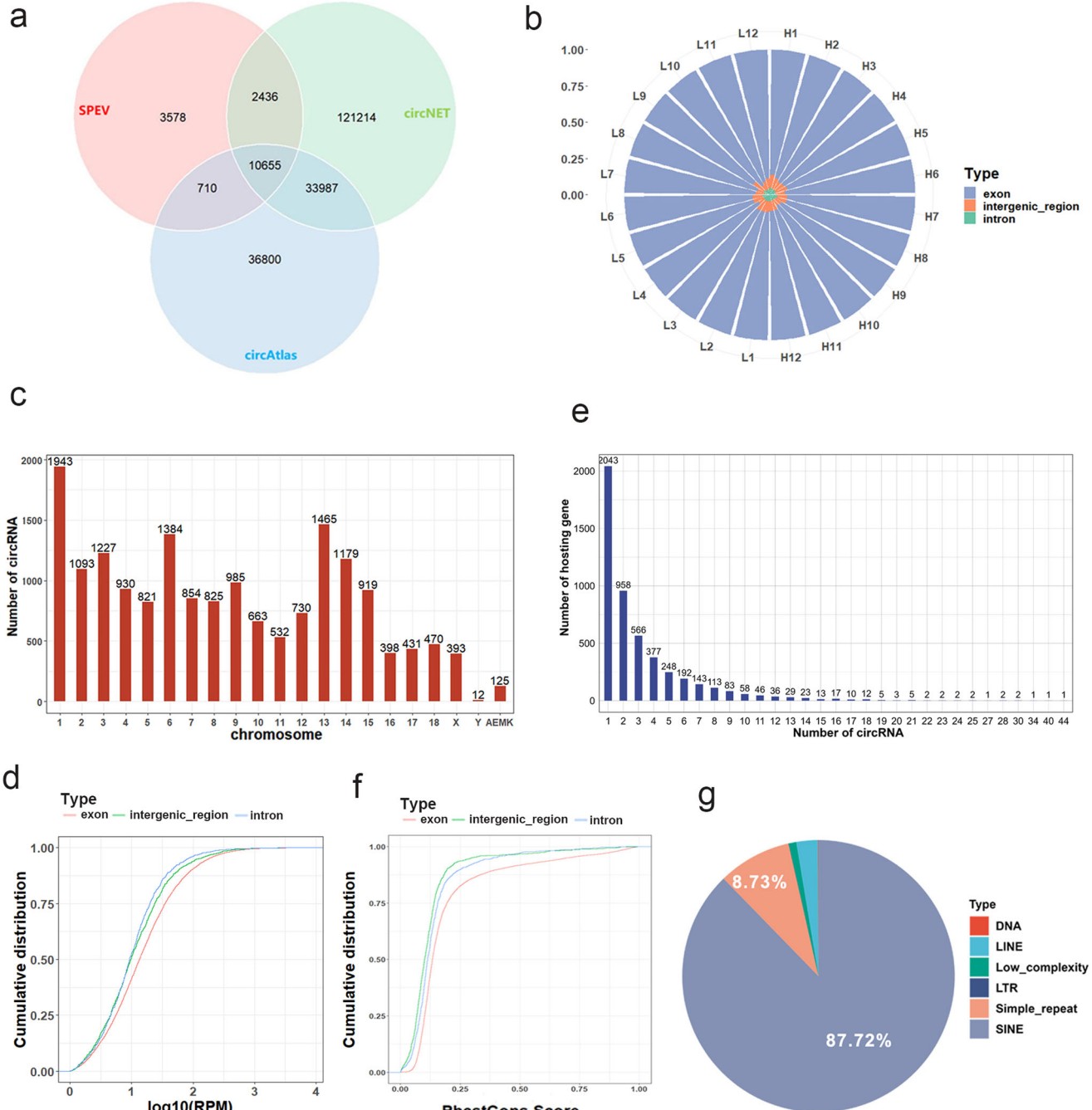

**Fig. 2 Identification and characterization of circRNAs in SPEVs. a** Venn diagram of circRNAs identified in the current study and porcine circRNAs contained in the circAtlas and circNET databases. **b** The proportions of exonic circRNAs, intronic circRNAs and intergenic region circRNAs in all samples. H represents high sperm motility group and L represents low sperm motility group. **c** The number of identified circRNAs in each chromosome. **d** Cumulative expression distribution of three different types of circRNAs. **e** Distribution of host genes encoding different numbers of circRNA isoforms. **f** Distribution of conservation scores of different types of circRNA. **g** The reverse complementary matches (RCMs) in the flanking sequences of back-splice junction (BSJ) sites of exonic circRNAs.

addition, we observed that miR-10384 and miR-143-3p target *MCL1*, *CREB1*, and *CREBBP*, which are involved in the PI3K-Akt signaling pathway. Thus, we further detected the expression of miR-10384 and miR-143-3p in SPEVs by quantitative PCR (qPCR) and found that both were significantly differentially expressed in SPEVs of the H group and L group (Fig. 4d).

**Circ-CREBBP is resistant to RNase R digestion.** Circ-CREBBP is highly conserved in many species, indicating that it may play an

important role in sperm function (Fig. 5a). To confirm whether the back-spliced sequences form a circular structure, we performed qPCR for circ-CREBBP in SPEVs treated with RNase R, which specifically degrades linear transcripts but not lariats or circRNAs. Circ-CREBBP was resistant to 0.2 U, 0.4 U, and 1 U RNase R digestion, suggesting that it is a closed-loop structure. In contrast, the corresponding linear cognate (*CREBBP*) and the housekeeping gene (*GAPDH*) were susceptible to exonucleolytic cleavage (Fig. 5b).

**Table 1 The 30 most expressed circRNAs.**

| circRNA ID | circRNA type | Host gene | Ensembl ID | Average expression |
|---|---|---|---|---|
| 5:22008784\|22009750 | Exon | PTGES3 | ENSSSCG00000000406 | 8176.69 |
| 12:11669306\|11671052 | Exon | PRKAR1A | ENSSSCG00000017259 | 6844.74 |
| 11:20710449\|20719541 | Exon | LRCH1 | ENSSSCG00000009408 | 5352.66 |
| 15:77234646\|77283482 | Exon | TLK1 | ENSSSCG00000015944 | 4653.14 |
| 5:2675422\|2692132 | Exon | TBC1D22A | ENSSSCG00000000955 | 4044.63 |
| 4:67679793\|67689219 | Exon | CSPP1 | ENSSSCG00000006202 | 3664.47 |
| 2:62307779\|62313757 | Exon | BRD4 | ENSSSCG00000022227 | 3365.23 |
| 10:32817449\|32827084 | Exon | UBAP2 | ENSSSCG00000010987 | 3296.04 |
| 17:13008092\|13031589 | Exon | PSD3 | ENSSSCG00000007034 | 3196.43 |
| 11:5503065\|5507316 | Exon | PAN3 | ENSSSCG00000009319 | 2807.65 |
| 17:13008092\|13012850 | Exon | PSD3 | ENSSSCG00000007034 | 2675.09 |
| 15:77126133\|77127562 | Exon | GORASP2 | ENSSSCG00000015943 | 2663.56 |
| X:115221683\|115223307 | Intergenic_region | – | – | 2511.73 |
| X:43052047\|43052598 | Exon | GRIPAP1 | ENSSSCG00000021106 | 2376.23 |
| 1:217786693\|217794582 | Intron | GLIS3 | ENSSSCG00000005224 | 2371.04 |
| 9:114937099\|114951665 | Exon | SUCO | ENSSSCG00000015485 | 2369.61 |
| 16:18379579\|18387297 | Exon | – | ENSSSCG00000031705 | 2223.63 |
| 9:66406846\|66409132 | Exon | NUCKS1 | ENSSSCG00000022398 | 2203.64 |
| AEMK02000682.1:1036980\|1040665 | Intergenic_region | – | – | 2156.69 |
| 11:1455412\|1463356 | Exon | MICU2 | ENSSSCG00000024667 | 2103.16 |
| 1:52134346\|52179177 | Exon | RIMS1 | ENSSSCG00000004280 | 2088.19 |
| 12:60348042\|60376920 | Exon | SLC5A10 | ENSSSCG00000018049 | 2062.68 |
| 7:65198472\|65198799 | Exon | SNX6 | ENSSSCG00000040912 | 1956.75 |
| 5:101790862\|101795355 | Exon | PAWR | ENSSSCG00000036560 | 1894.88 |
| 18:54260019\|54270584 | Exon | SUGCT | ENSSSCG00000035581 | 1886.17 |
| 10:8265577\|8271193 | Exon | RRP15 | ENSSSCG00000027846 | 1879.99 |
| 8:87799383\|87820356 | Exon | ELF2 | ENSSSCG00000029920 | 1829.12 |
| 14:40098524\|40101523 | Exon | BICDL1 | ENSSSCG00000009896 | 1758.25 |
| 11:1141224\|1141776 | Exon | XPO4 | ENSSSCG00000009276 | 1642.82 |
| 18:54393586\|54404485 | Exon | CDK13 | ENSSSCG00000016769 | 1544.60 |

**Circ-CREBBP improved sperm motility and inhibited sperm apoptosis.** To investigate the biological function of circ-CREBBP in sperm motility, we first detected the expression of circ-CREBBP in sperm of the H group and L group. The results showed that the expression of circ-CREBBP was significantly higher in H group than L group (Fig. 5c), which was consistent with its expression trend in SPEVs. Then, we designed circ-CREBBP siRNAs against the back-spliced sequence of circ-CREBBP (Fig. 5d). H-SPEVs electroporated with circ-CREBBP siRNA were incubated with sperm for 96 hours, and the sperm motility was detected at different time points. The results showed that Si-circ-CREBBP significantly decreased sperm motility compared with the control group from 12 h to 96 h (Fig. 5e). At the same time, we performed a qPCR assay to detect the expression of circ-CREBBP in the sperm of the two groups at 48 h and 96 h. The results showed that circ-CREBBP in sperm incubated with SPEV electroporated with siRNA was successfully silenced (Fig. 5f). These data suggested that low expression of circ-CREBBP in sperm leads to a significant decrease in sperm motility.

To further explore the effect of circ-CREBBP on spermatozoa, the spermatozoa of the Si-circ-CREBBP group and control group were stained with Annexin V-FITC and PI and then analyzed by flow cytometry. The results suggested that the percentage of apoptotic sperm increased significantly in the Si-circ-CREBBP group (Fig. 5g). Compared with the control group, the average percentage of surviving cells in the Si-circ-CREBBP group decreased by 4.00% and 3.20% on day 2 and day 4, respectively. These data showed that the number of apoptotic sperm significantly increased in the Si-circ-CREBBP group ($P$ value < 0.05), indicating that circ-CREBBP inhibits sperm apoptosis.

Furthermore, we assessed the levels of ATP in spermatozoa incubated with SPEV electroporated with circ-CREBBP siRNA. We observed that Si-circ-CREBBP induced a decrease in cellular ATP levels. As shown in Fig. 5h, the ATP levels of spermatozoa for 48 h and 96 h in the control group were 11.48 nmol/mg and 5.02 nmol/mg, respectively. In contrast, the ATP levels of spermatozoa treated with Si-circ-CREBBP for 48 h and 96 h decreased to 8.48 nmol/mg and 4.70 nmol/mg, respectively. These results suggested that the levels of ATP in spermatozoa were markedly decreased ($P$ value < 0.01) in the Si-circ-CREBBP group, which further confirmed that the number of apoptotic sperm increased after the decrease in circ-CREBBP expression.

**Circ-CREBBP targets miR-10384 and miR-143-3p to downregulate their expression.** CircRNAs may act as miRNA sponges for multiple miRNAs. To identify the underlying molecular mechanism by which circ-CREBBP inhibits sperm apoptosis, we searched for the potential miRNA targets of circ-CREBBP using RNAhybrid and miRanda. The results showed that circ-CREBBP contains complementary binding sites with miR-10384 and miR-143-3p (Fig. 6a). Then, the circ-CREBBP luciferase reporter vectors were transfected with miRNA mimics (miR-10384 and miR-143-3p) or inhibitors (miR-10384 and miR-143-3p) into 293 T cells. Luciferase expression effectively decreased upon coexpression of miR-10384 mimic or miR-143-3p mimic with circ-CREBBP-WT (Fig. 6b). To reverse verify the targeting relationship, circ-CREBBP-MUT1, which introduced mutations into the seed sequences of predicted miR-10384 binding sites, and circ-CREBBP-MUT2, which introduced mutations into the seed sequences of predicted miR-143-3p binding sites, were cotransfected with miRNA mimics or inhibitors into 293 T cells. The results showed that both the miRNA mimic and inhibitor had no effect on circ-CREBBP-MUTs (Fig. 6b), indicating that circ-CREBBP could specifically bind miR-10384 and miR-143-3p.

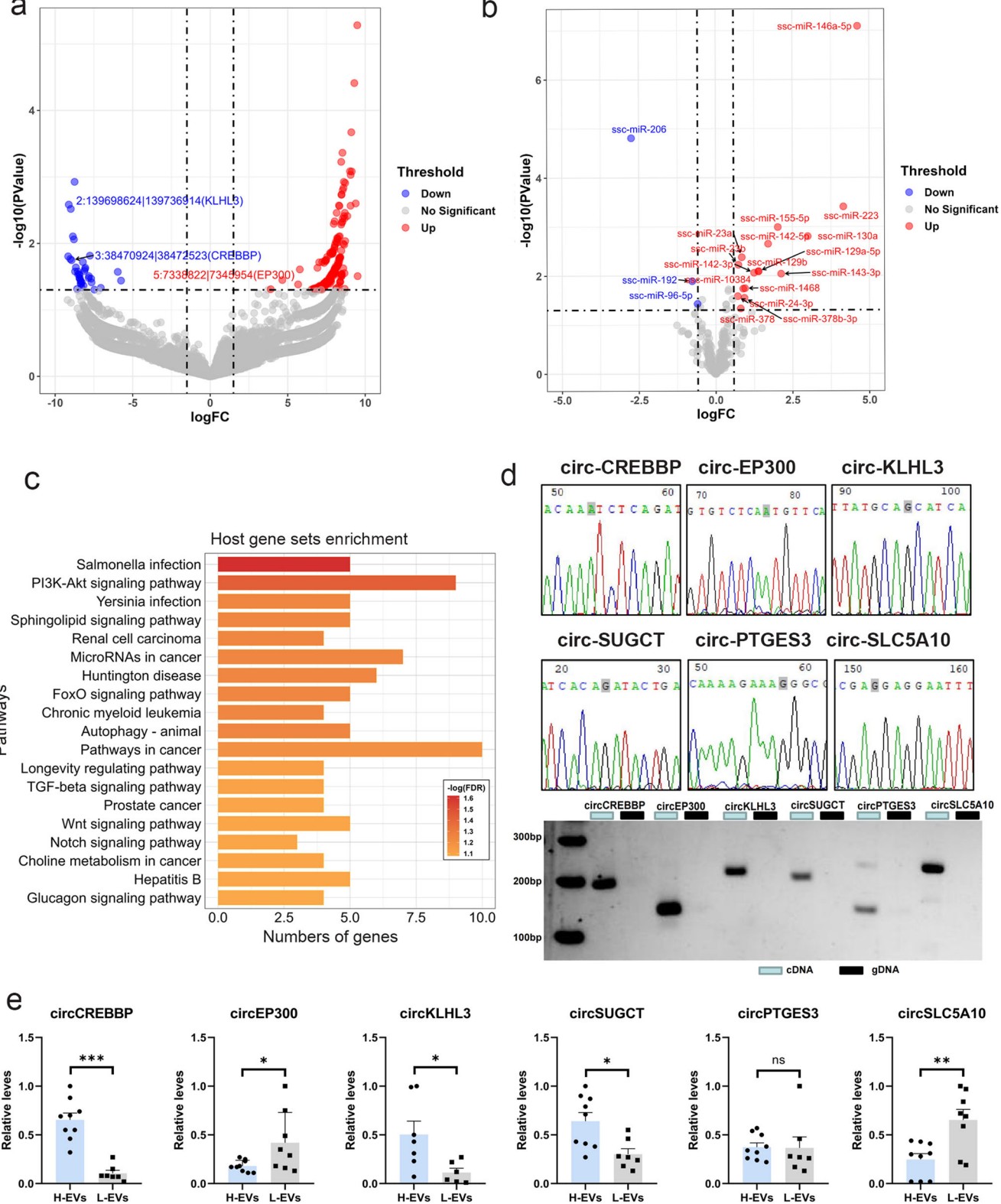

**Fig. 3 Whole-transcriptome profiles of the H-SPEV and L-SPEV groups. a** Volcano plot of DECs. Red represents upregulation in the L group, and blue represents upregulation in the H group. **b** Volcano plot of 19 DEMs. Red represents upregulation in the L group, and blue represents upregulation in the H group. **c** KEGG pathway analysis of host genes of DECs. **d** The back-splice junction sites of six circRNAs were validated by qPCR followed by Sanger sequencing. The junction sites of circRNAs are marked in gray. **e** qPCR validation results for six circRNAs in the two groups. Data are presented as the mean ± SEM. *$P \leq 0.05$, **$P \leq 0.01$, ***$P \leq 0.001$ (Student's *t* test), $n = 3$ for each group.

**Table 2 The top 20 differentially expressed circRNAs.**

| circRNA ID | Host gene | Ensembl ID | LogCPM | LogFC | P-value | Regulated |
|---|---|---|---|---|---|---|
| 16:43997828\|44055338 | – | n/a | 6.87 | −9.16 | 0.02 | Down |
| 2:139698624\|139736914 | KLHL3 | ENSSSCG00000014321 | 6.83 | −9.12 | 2.61E-03 | Down |
| 8:96833216\|96845587 | LARP1B | ENSSSCG00000034280 | 6.74 | −9.03 | 0.02 | Down |
| 4:2792392\|2810920 | PTK2 | ENSSSCG00000038397 | 6.71 | −8.99 | 3.03E-03 | Down |
| 3:38470924\|38472523 | CREBBP | ENSSSCG00000007951 | 6.66 | −8.94 | 0.02 | Down |
| 3:6981006\|6988833 | CEP20 | ENSSSCG00000033377 | 6.58 | −8.86 | 0.01 | Down |
| 11:4615097\|4652306 | USP12 | ENSSSCG00000009302 | 6.52 | −8.79 | 0.01 | Down |
| 18:38826704\|38843265 | DPY19L1 | ENSSSCG00000027063 | 6.48 | −8.75 | 1.19E-03 | Down |
| 14:85924102\|85930515 | – | n/a | 6.45 | −8.72 | 0.03 | Down |
| 10:15051106\|15053287 | AHCTF1 | ENSSSCG00000010863 | 6.40 | −8.67 | 0.02 | Down |
| 14:54638310\|54639066 | MTR | ENSSSCG00000010143 | 6.57 | 8.85 | 3.88E-03 | Up |
| 6:18553565\|18553968 | – | ENSSSCG00000025417 | 6.73 | 9.02 | 8.23E-04 | Up |
| 5:77014931\|77024824 | – | ENSSSCG00000000807 | 6.76 | 9.05 | 2.72E-03 | Up |
| 15:76579909\|76608202 | MYO3B | ENSSSCG00000032177 | 6.77 | 9.06 | 9.34E-04 | Up |
| 4:118066785\|118069700 | – | ENSSSCG00000006870 | 6.81 | 9.10 | 2.13E-04 | Up |
| 14:71852747\|71853159 | CCAR1 | ENSSSCG00000010243 | 6.83 | 9.13 | 8.28E-04 | Up |
| 5:68310430\|68321820 | ERC1 | ENSSSCG00000000755 | 6.99 | 9.30 | 3.88E-05 | Up |
| 13:204825458\|204826392 | MX2 | ENSSSCG00000012076 | 7.08 | 9.39 | 2.51E-03 | Up |
| 17:43759621\|43769684 | TOP1 | ENSSSCG00000007355 | 7.17 | 9.49 | 5.23E-06 | Up |
| 13:134233747\|134234461 | – | ENSSSCG00000011850 | 7.20 | 9.51 | 0.03 | Up |

Subsequently, the expression of miR-10384 and miR-143-3p was detected in sperm incubated with SPEVs electroporated with Si-circ-CREBBP on day 2 and day 4 using qPCR. The results showed that the expression of miR-10384 and miR-143-3p significantly increased after sperm were incubated with Si-circ-CREBBP (Fig. 6c). These data demonstrated that circ-CREBBP has a strong targeting relationship with miR-10384 and miR-143-3p and regulates their expression.

**MCL1, CREBBP, and CREB1 are the effectors of miR-10384 and miR-143-3p**. According to miRbase, the sequences of miR-143-3p are highly conserved in a variety of species (http://www.mirbase.org/), including pig, human, mouse (Fig. 6d). To further identify downstream effectors of miR-10384 and miR-143-3p, we used RNAhybrid to identify potential target genes of miR-10384 and miR-143-3p. The 3′UTRs of MCL1, CREBBP and CREB1 were found to contain a miR-10384 or miR-143-3p target-binding site (Fig. 6e). Then, their 3′UTRs were cloned into luciferase reporter vectors, and the luciferase reporter vectors were cotransfected with miR-10384 and miR-143-3p mimic or inhibitor into 293 T cells. The results showed that luciferase expression effectively decreased upon coexpression of miR-10384 with the MCL1 3′UTR and miR-143-3p with the CREBBP 3′UTR and CREB1 3′UTR (Fig. 6e). Moreover, mutations were introduced into the seed sequences of predicted miR-10384 binding sites and miR-143-3p binding sites. The results showed that the mutations within the MCL1 3′UTR abolished the inhibitory effects of miR-10384 on luciferase expression and that the mutations within the CREBBP 3′UTR and CREB1 3′UTR abolished the inhibitory effects of miR-143-3p on luciferase expression (Fig. 6e).

Moreover, we detected the expression of miR-10384 in sperm incubated with SPEVs electroporated with miR-10384 mimic or inhibitor on day 2 and day 4 using qPCR. The results showed that the miR-10384 mimic or inhibitor was successfully transferred to SPEVs. Compared with the control group (SPEVs without electroporation), the expression of miR-10384 significantly increased in sperm incubated with SPEVs electroporated with miR-10384 mimic and significantly decreased in sperm incubated with SPEVs electroporated with miR-10384 inhibitor (Fig. 6f). Similarly, we also examined the expression of miR-143-3p in sperm incubated with SPEVs electroporated with miR-143-3p mimic or inhibitor. Quantitative PCR analysis suggested that the

expression of miR-143-3p was significantly increased in the mimic group (Fig. 6f). Accordingly, we detected the expression of MCL1 and CREBBP in each group. The results showed that the mRNA expression of MCL1 was significantly increased in sperm incubated with miR-10384 inhibitor on day 2 and day 4 and significantly decreased in sperm incubated with miR-10384 mimic (Fig. 6g). Similarly, the expression levels of CREBBP and CREB1 significantly increased in sperm incubated with miR-143-3p inhibitor on day 2 and day 4 and decreased in the mimic group (Fig. 6g). These results demonstrated that MCL1, CREBBP, and CREB1 are directly regulated by miR-10384 and miR-143-3p, respectively.

**MCL1, CREBBP, and CREB1 inhibit sperm apoptosis by regulating the PI3K-Akt signaling pathway**. MCL1, an anti-apoptotic member of the BCL2 family, regulates apoptosis by interacting with many other apoptosis regulators. We examined the protein expression of MCL1 in sperm on day 2 and found that the protein expression of MCL1 was markedly decreased in sperm incubated with the miR-10384 mimic (Fig. 7a and Supplementary Fig. 2). Previous studies have shown that the degradation of the MCL1 protein occurs mainly through the ubiquitination pathway. Therefore, we detected the ubiquitination level of MCL1 using immunoprecipitation experiments. The results showed that the ubiquitination level of MCL1 increased in sperm incubated with miR-10384 mimic (Fig. 7a and Supplementary Fig. 2). In contrast, the ubiquitination level of MCL1 in sperm incubated with miR-10384 inhibitor was lower than that in the miR-10384 mimic group and control group. Since MCL1 inhibits cell apoptosis by binding with BAX, we performed co-immunoprecipitation (Co-IP) and immunofluorescence assays to detect the interactions between MCL1 and BAX in sperm. As shown in Fig. 7a (Supplementary Fig. 2), BAX was present in the MCL1 precipitates, and the amount of BAX binding to MCL1 in the miR-10384 mimic group was obviously lower than that in the miR-10384 inhibitor group. The results suggested that the degradation of MCL1 resulted in a decrease in the binding of BAX, a proapoptotic member. We further detected the total expression of BAX in each group. Western blot analysis results showed that the protein expression level of BAX was higher in sperm incubated with miR-10384 mimic than in the miR-10384 inhibitor group

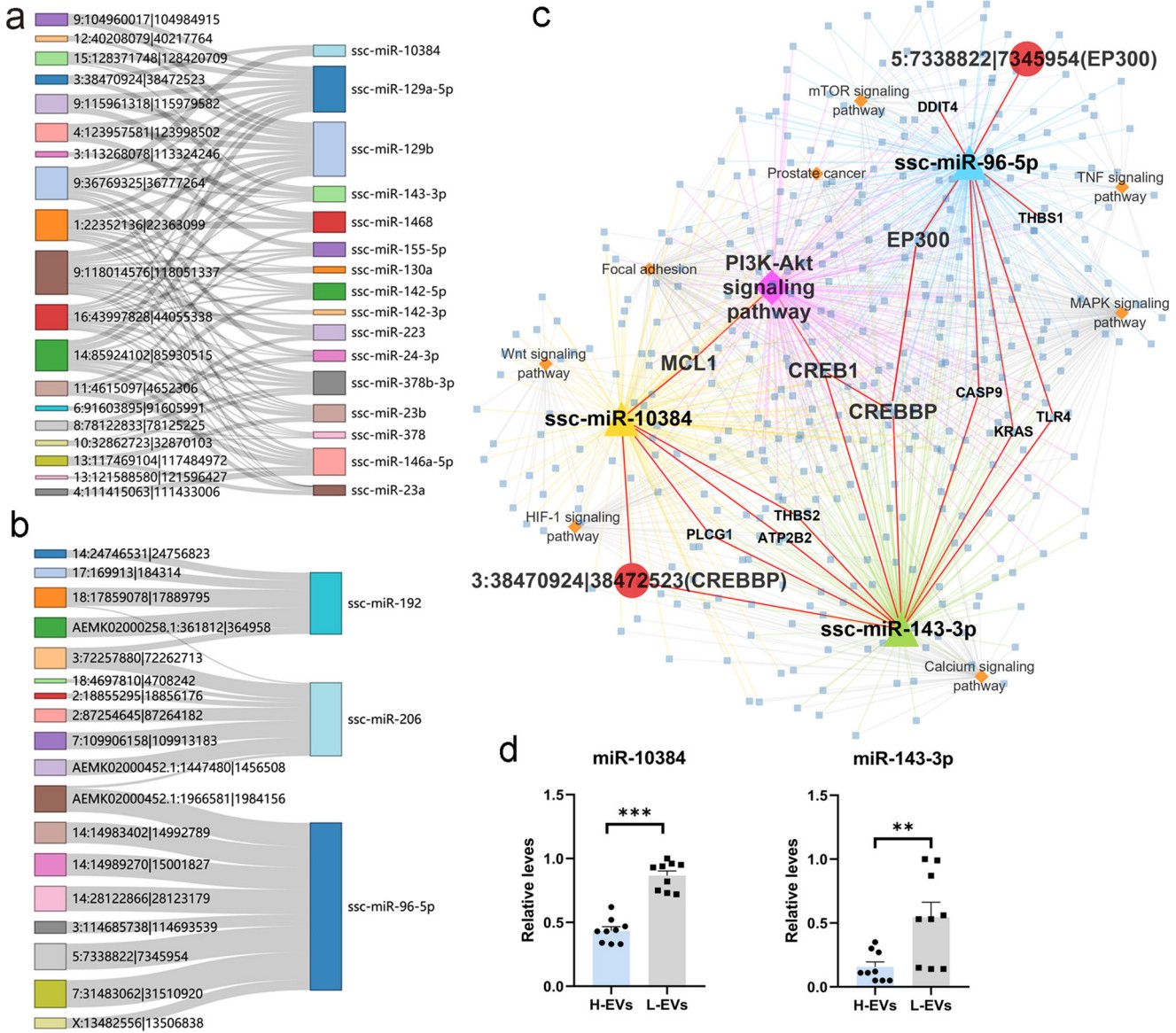

**Fig. 4 Regulatory networks of DEC-DEM-genes. a** The regulatory network of upregulated circRNAs and downregulated target miRNAs in H group. **b** The regulatory network of downregulated circRNAs and upregulated target miRNAs in H group. **c** A network of two important circRNAs, their target DEMs, target genes and related pathways. **d** The relative expression of miR-10384 and miR-143-3p in H-SPEVs and L-SPEVs was detected by qPCR. **\*\***$P \leq 0.01$, **\*\*\***$P \leq 0.001$ (Student's *t* test), $n = 3$ for each group.

(Fig. 7a and Supplementary Fig. 2). Immunofluorescence data indicated that MCL1 and BAX colocalized in the cytoplasm and tails of sperm, which provided new evidence for the interaction of the two proteins (Fig. 7b). Finally, we detected the expression of CASP3 and found that the protein expression of CASP3 increased markedly in the miR-10384 mimic group, suggesting that miR-10384 can promote sperm apoptosis. In contrast, the expression of CASP3 in the inhibitor group was very low.

Additionally, we demonstrated that miR-143-3p regulated the mRNA expression of *CREBBP* and *CREB1*. We detected the protein expression of CREBBP and CREB1 in sperm on day 2 and found that CREBBP and CREB1 were decreased in sperm incubated with miR-143-3p mimic (Fig. 7a and Supplementary Fig. 3). CREBBP, as a transcription cofactor, can combine with CREB1 to promote BCL2 expression. After activation, CREB1 binds to the promoters of *BCL2* and *MCL1* and activates *BCL2* and *MCL1* expression, protecting cells from undergoing programmed cell death[36]. Therefore, we performed Co-IP assays to

detect the interactions between CREBBP and CREB1 in sperm. We observed that CREB1 was present in the CREBBP precipitates, and the amount of CREB1 binding to CREBBP in the miR-143-3p mimic group was lower than that in the miR-143-3p inhibitor group (Fig. 7a and Supplementary Fig. 3). In addition, phosphorylation of CREB1 at Ser-133 was detected by western blot. The results showed that the expression of CREB1 phosphorylation also decreased in the miR-143-3p mimic group (Fig. 7a and Supplementary Fig. 3). Then, we detected the protein expression of MCL1 and BCL2 in different groups. We observed that the expression levels of MCL1 and BCL2 in sperm incubated with miR-143-3p mimic were lower than those in sperm incubated with inhibitor (Fig. 7a and Supplementary Fig. 3). Furthermore, the ubiquitination level of MCL1 in sperm incubated with miR-143-3p inhibitor was obviously lower than that in the miR-143-3p mimic group (Fig. 7a and Supplementary Fig. 3). Moreover, BAX combined with BCL2 was markedly decreased in the miR-143-3p mimic group (Fig. 7a and

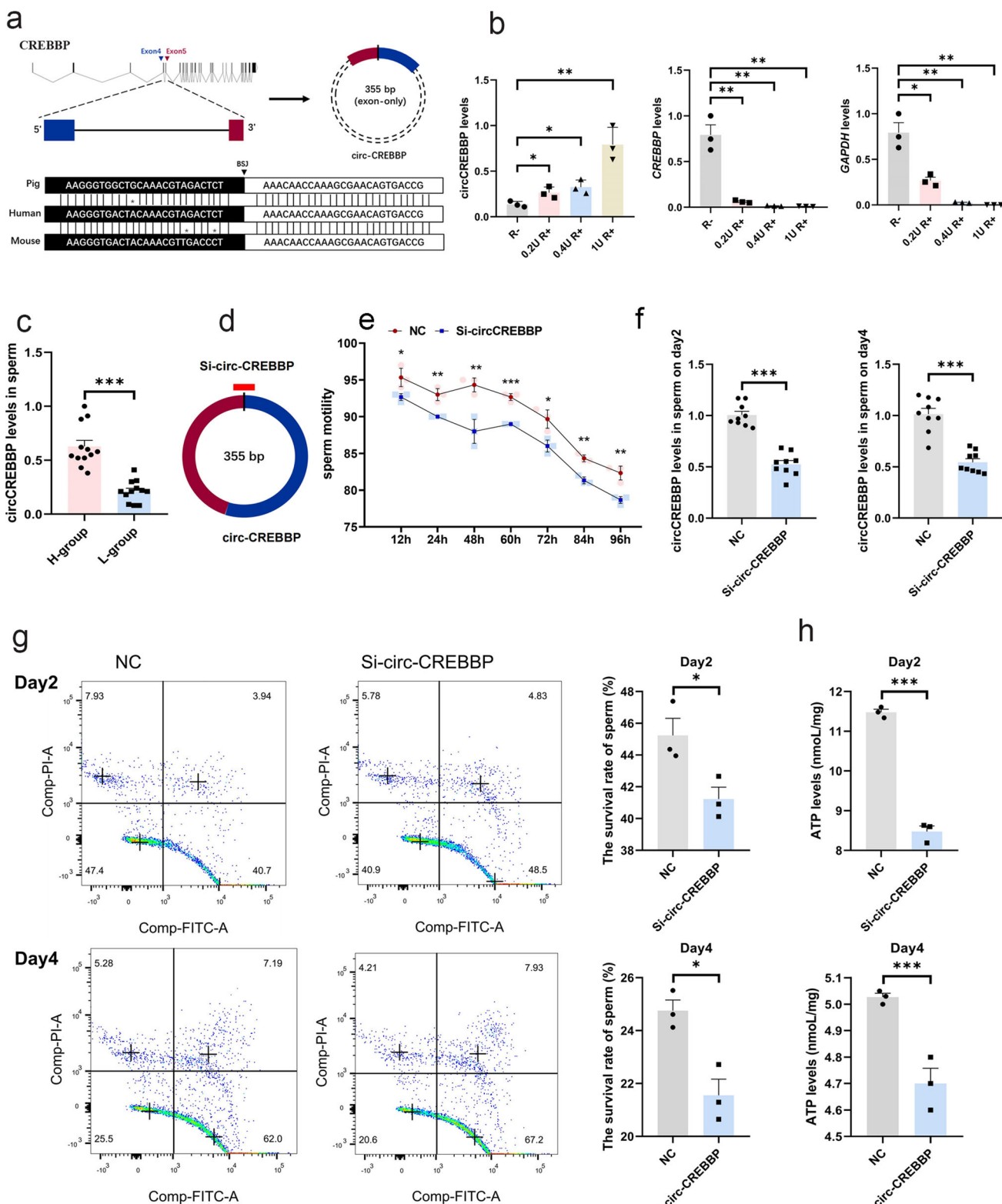

**Fig. 5 Effect of circ-CREBBP on sperm in vitro. a** Upper panel: Schematic representation of circ-CREBBP formation. Lower panel: The sequence of circ-CREBBP is conserved in humans and mice. **b** Compared to *CREBBP* mRNA and *GAPDH* mRNA, qPCR revealed that circ-CREBBP was resistant to RNase R. **c** The relative expression levels of circ-CREBBP in sperm of the H group and L group were detected by qPCR. **d** Schematic diagram of Si-circ-CREBBP. **e** The total motility of sperm incubated with EV (NC) and EV electroporated with Si-circ-CREBBP from 12 h to 96 h, respectively. **f** The expression of circ-CREBBP in sperm coincubated with EV (NC) or EV-Si-circ-CREBBP for two days and four days was determined by qPCR. **g** The survival rate in the NC and EV-Si-circ-CREBBP groups for two days and four days, respectively. **h** The cellular ATP levels in sperm coincubated with EV (NC) and EV-Si-circ-CREBBP for two days and four days. (**b**, **c**, **e**, **f**, **h**) All data are presented as the mean ± SEM. *$P ≤ 0.05$, **$P ≤ 0.01$, ***$P ≤ 0.001$ (ANOVA or Student's *t* test). (**b**, **e**, **f**, **h**) $n = 3$ for each group; (**c**) $n = 4$ for each group.

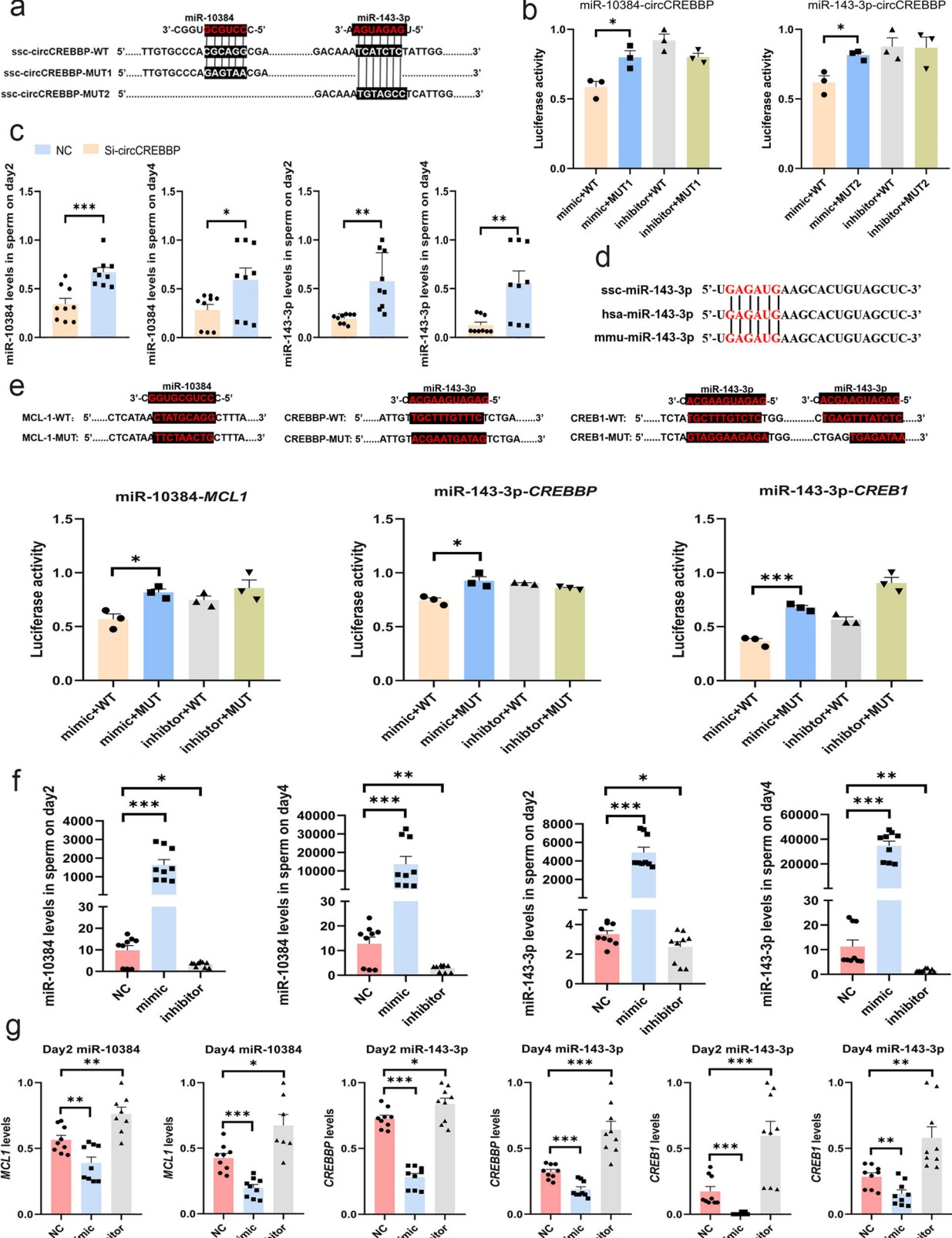

Supplementary Fig. 3). Immunofluorescence data showed that CREBBP and CREB1 colocalized in the tails of sperm (Fig. 7b). Interestingly, we found that CREB1 was only expressed in the post acrosomal ring of sperm and sperm tails. In particular, CREB1 was highly expressed in the post acrosomal ring of sperm, which is considered to act as a diffusion barrier to segregate important molecules required for fertilization within specific domains. Finally, we detected the expression levels of BAX and CASP3 in each group. The results showed that BAX and CASP3 expression was significantly increased in sperm incubated with miR-143-3p mimic, indicating that high expression of miR-143-3p can promote sperm apoptosis.

**Fig. 6 Circ-CREBBP targets miR-10384 and miR-143-3p. a** The predicted target-binding sites between circ-CREBBP and miR-10384 and miR-143-3p. **b** Luciferase activity assays showed that circ-CREBBP targets miR-10384 and miR-143-3p. **c** The relative expression of miR-10384 and miR-143-3p in sperm coincubated with EV (NC) and Si-circ-CREBBP for two days and four days, respectively. **d** The sequence of miR-143-3p is highly conserved in different species. **e** The target-binding sites between miR-10384 and *MCL1* and between miR-143-3p and *CREBBP* and *CREB1* are shown. Luciferase activity assays showed that miR-10384 targets *MCL1* and miR-143-3p targets *CREBBP* and *CREB1*, respectively. **f** The relative expression of miR-10384 and miR-143-3p in sperm incubated with EV (NC), EV-miRNA mimic and EV-miRNA inhibitor for two days and four days was determined by qPCR. **g** The mRNA expression of *MCL1, CREBBP* and *CREB1* in sperm coincubated with EV (NC), EV-miR10384/EV-miR-143-3p mimic and EV-miR-10384/EV-miR-143-3p inhibitor for two days and four days was determined by qPCR. (**b, c, e–g**) Data are presented as the mean ± SEM. *$P \leq 0.05$, **$P \leq 0.01$, ***$P \leq 0.001$ (ANOVA or Student's *t* test), $n = 3$ for each group.

**Circ-CREBBP regulates the expression of MCL1, CREBBP, and CREB1.** To verify whether circ-CREBBP indirectly regulates the expression of MCL1, CREBBP and CREB1, we detected the mRNA and protein expression of MCL1, CREBBP, and CREB1 in sperm incubated with Si-circ-CREBBP. The results showed that the mRNA expression of MCL1, CREBBP, and CREB1 significantly decreased in sperm with circ-CREBBP knockdown on day 2 and 4, respectively (Fig. 7c). In addition, western blot analysis results showed that CREBBP, CREB1, p-CREB1, and MCL1 markedly decreased in sperm of the Si-circ-CREBBP group (Fig. 7d and Supplementary Fig. 4). Next, we detected the expression of BCL2 and CASP3. The results showed that BCL2 expression was decreased and CASP3 expression was increased in the Si-circ-CREBBP group. In addition, Co-IP assays showed that CREB1 combined with CREBBP was decreased in the Si-circ-CREBBP group, and CREB1 phosphorylation was also decreased in the Si-circ-CREBBP group (Fig. 7d and Supplementary Fig. 4). Furthermore, the ubiquitination level of MCL1 in the Si-circ-CREBBP group was markedly higher than that in the control group (Fig. 7d and Supplementary Fig. 4). We also observed that BAX binding to MCL1 and BCL2 decreased obviously in the Si-circ-CREBBP group (Fig. 7d and Supplementary Fig. 4). The above results suggested that circ-CREBBP indirectly regulated the expression of MCL1, CREBBP, and CERB1 through miR-10384 and miR-143-3p.

**Discussion**

Pigs have become an ideal model for human biomedical research because of their high similarity to humans in terms of genome sequences, anatomy and physiology. Human reproductive diseases are becoming increasingly prominent, such as infertility. Sperm motility provides an objective measurement of semen quality. At present, the swine industry is recording the semen quality of each boar used for artificial insemination, including sperm motility and other parameters. Thus, data on boar reproduction can be used to study the molecular basis of semen quality. The research results of pig semen quality can provide valuable information for male reproductive diseases.

Extracellular vesicles widely exist in animal body fluids. In recent years, it has been found that SPEVs play an important role in sperm motility and can affect sperm maturation, capacitation and egg binding. Some studies have shown that it is expected to become a new molecular marker for disease diagnosis (such as prostate cancer, azoospermia, asthenospermia, etc.) and prognosis evaluation. CircRNAs are a class of noncoding RNA molecules without a 5′-end cap and a 3′-end poly(A) tail. It has been reported that circRNAs participate in many important biological functions, such as apoptosis and oxidative stress. Therefore, we comprehensively analyzed circRNAs in SPEVs from boars with high or low sperm motility and studied their function.

In the current study, our data identified a total of 17,379 circRNAs, of which 80% can be found in the database. This result suggested that these circRNAs are relatively conserved and stable

in mammals and may play an important role in biological processes. Compared with data previously reported in other tissues of pigs[37], humans[12] and rats[38], circRNAs in SPEVs showed similar characteristics, such as the proportion of exonic, intronic and intergenic circRNAs. Our results showed that circRNAs in the SPEVs were mainly derived from exons, accounting for 84.9%. This result is consistent with a study on human circRNAs[29]. In addition, we found that circRNAs in SPEVs were distributed on all chromosomes, and the number of circRNAs was the highest on chromosome 1[39]. Interestingly, we found 125 circRNAs that were derived from mitochondria for the first time, and these circRNAs may play an important role in sperm motility.

We investigated the functional relevance of SPEV circRNAs under the hypothesis that their function is associated with the host gene. In our study, 24 genes harbored 20 or more circRNA isoforms. Some of these circRNA host genes, such as *KMT2C*[40], *BRWD1*[41,42], and *CHD2*[43], have been implicated in sperm function and male fertility. In addition, some of these host genes have been used as important prostate cancer marker genes, such as *TRIM24*[44], *UBAP2*[45], *ULK4*[46], and *SMARCC1*[47]. Interestingly, we found that circRNAs derived from these host genes in SPEVs were also identified in sperm. For example, *SLC5A10* produced 21 circRNA isoforms. Godia et al. found *SLC5A10* with 5 circRNA isoforms in pig sperm. In addition, *SLC5A10* hosted a circRNA with a significant abundance correlation with sperm motility[48]. The authors also reported that ssc-circ-0345 from *SLC5A10* can regulate miR-423-5p and let-7c, which were altered in patients with oligospermia and severe asthenospermia[48]. Similarly, our results showed that 11 circ-SLC5A10 isoforms can target miR-423-5p.

In the current study, a total of 222 differentially expressed circRNAs were identified, most of them were more highly expressed in the low sperm motility group. We performed enrichment analysis of the host genes of DECs and found that many genes were related to sperm motility, such as *CREBBP*, *EP300*, *PTK2*, etc. CREBBP may play a role in azoospermia[49]. In the current study, two circRNAs derived from *PTK2* and *CREBBP* belonged to the top 5 circRNAs with the largest differential expression between groups. A recent study showed that *PTK2* hosted a circRNA with a significant abundance correlation with sperm motility[48]. In our study, we found that circ-PTK2 was highly expressed in SPEVs of boars with low sperm motility. It has been reported that the expression of circ-PTK2 is increased in colorectal cancer tissues and is positively correlated with tumor growth, metastasis, and poor survival rates[50].

In our previous studies, we explored differentially expressed miRNAs associated with sperm motility in SPEVs[24,25]. Some important DEMs were identified again, such as ssc-miR-223, ssc-miR-23a, ssc-miR-23b, ssc-miR-24-3p, ssc-miR-378, and ssc-miR-155-5p. To elucidate the functional relevance of circRNAs as miRNA sponges, we established a DECs-DEMs interaction network and found that 37 circRNAs and 19 miRNAs have ceRNA relationships. Remarkably, circ-CREBBP and circ-EP300 were linked to ssc-miR-10384, ssc-miR-143-3p, and ssc-miR-96-5p.

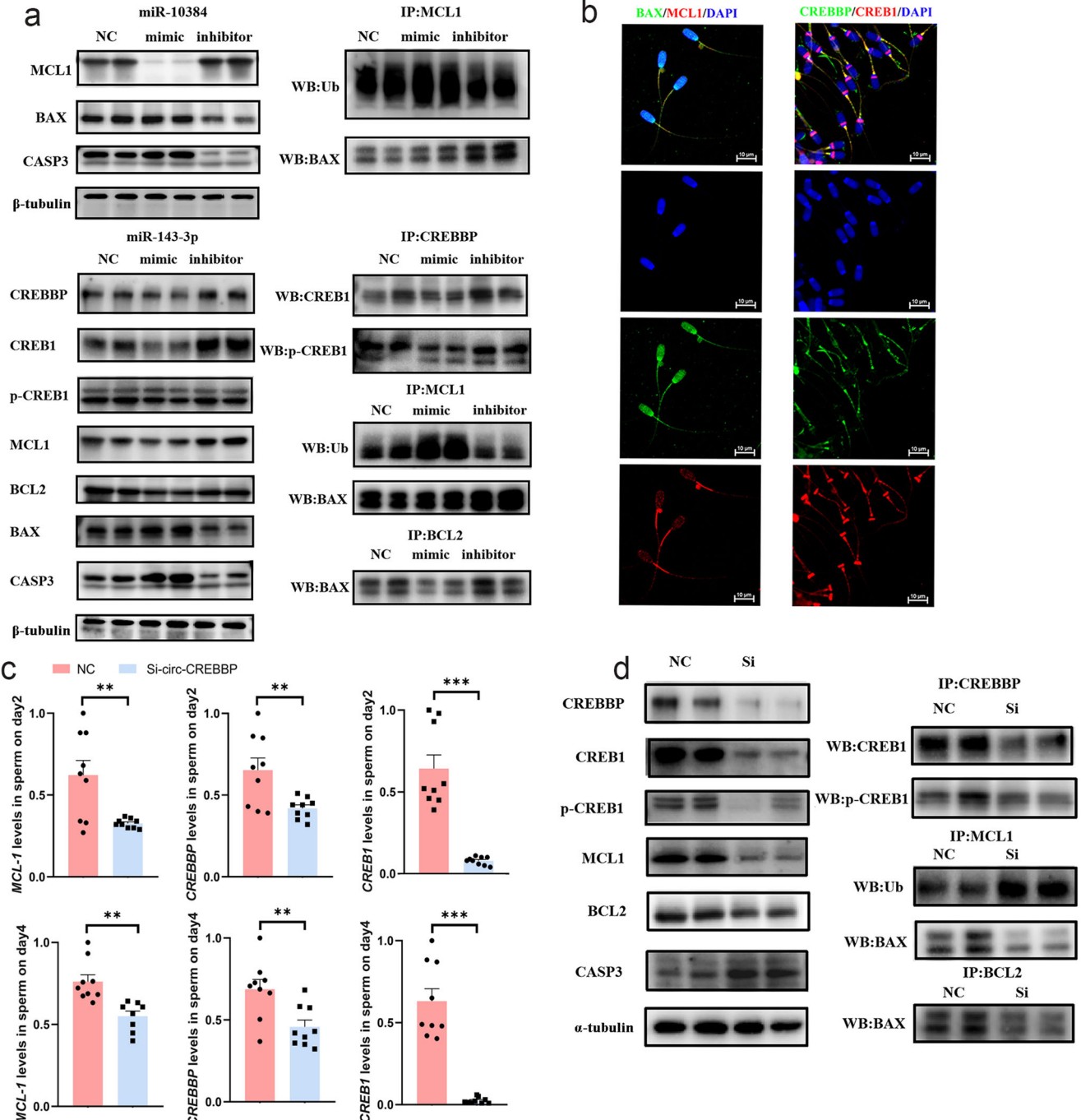

**Fig. 7 Circ-CREBBP maintains the survival of sperm via the PI3K/Akt pathway. a** Western blot assays were used to evaluate the total protein levels of PI3K/Akt and apoptosis pathway members MCL1, CREBBP, CREB1, BCL2, BAX, and CASP3, the ubiquitination level of MCL1 and the phosphorylation levels of CREB1 in sperm incubated with EV, EV-miR-10384/EV-miR-143-3p mimic, and EV-miR-10384/EV-miR-143-3p inhibitor. **b** Immunofluorescence data showed that MCL1 and BAX and CREBBP and CREB1 were mainly colocalized in sperm tails. **c** The mRNA expression of *MCL1*, *CREBBP* and *CREB1* in sperm incubated with EV (NC) or Si-circ-CREBBP was detected by qPCR. Data are presented as the mean ± SEM. *$P \leq 0.05$, **$P \leq 0.01$, ***$P \leq 0.001$ (Student's $t$ test), $n = 3$ for each group. **d** Western blot assays were used to detect the protein levels of PI3K/Akt and apoptosis pathway members MCL1, CREBBP, CREB1, BCL2, BAX, and CASP3, the phosphorylation levels of CREB1 and the ubiquitination level of MCL1 in sperm incubated with EV (NC) or EV-Si-circ-CREBBP.

CREBBP is highly expressed in spermatogonia and early spermatocytes and plays an important role in regulating sperm energy metabolism[51]. In addition, it has been reported that p300 plays an essential role in the cellular processes of growth and differentiation[52]. Bao et al. found that p300 is recruited through ACT to activate the testis-specific transcription factor CREM in the early stage of haploid spermatids[53]. Boussouar et al. found

that a specific CREBBP/p300-dependent gene expression program drives metabolic remodeling in late stages of spermatogenesis[51]. In addition, CREBBP/p300 proteins have been found to be expressed in Sertoli cells, which are responsible for the maintenance of spermatogenesis in adults. In addition, transferrin is one of the main secretory products expressed by differentiated Sertoli cells. CREBBP/p300 has been found to be

involved in regulating the activation of the transferrin promoter[54]. Furthermore, a large amount of evidence has shown that CREBBP/p300 functions as an interactor with CREB1[54,55].

To understand the effect of circ-CREBBP on sperm motility and function, spermatozoa were incubated with SPEVs electroporated with Si-circ-CREBBP. We observed that sperm motility significantly decreased and the percentage of apoptotic sperm significantly increased. At the same time, the levels of ATP in spermatozoa incubated with SPEV electroporated with circ-CREBBP siRNA were also decreased. These data suggested that circ-CREBBP plays an important role in maintaining sperm motility and inhibiting sperm apoptosis. We verified that circ-CREBBP targeted miR-10384 and miR-143-3p and indirectly regulated the mRNA and protein expression of MCL1, CREBBP, and CREB1. MCL1 is an antiapoptotic member of the BCL-2 family. Overexpression of MCL1 delays apoptosis induced by c-Myc overexpression and other cytotoxic drugs[56,57]. Our results showed that the MCL1 level was significantly decreased in sperm with Si-circ-CREBBP. In addition, MCL1 is induced by CREB[58], a transcription factor that regulates cell proliferation, survival and differentiation[59]. CREB can promote B-cell survival via its induction of MCL1[60]. Inhibition of CREB expression resulted in decreases in both MCL1 levels and viability[60]. In this study, our results suggested that the expression level of CREB1 was also significantly reduced in sperm with Si-circ-CREBBP. Moreover, CREB1 has been reported to activate target gene expression by promoting the recruitment of CREBBP[61]. Parker et al. reported that phosphorylation of CREB at Ser-133 induces complex formation with CREB-binding protein via a direct mechanism[62]. Upon phosphorylation at Ser 133, p-CREB1 can activate transcriptional regulation through its interaction with CREBBP/p300[55]. Therefore, we detected the expression of CREB1 phosphorylation in each group and found that pCREB1 also decreased in the miR-143-3p mimic group and Si-circ-CREBBP group. Similarly, activated CREB can also bind to the BCL2 promoter[36]. Our results showed that BCL2 expression was significantly decreased in sperm of the Si-circ-CREBBP group. Likewise, the CREBBP expression level was significantly decreased in sperm of the Si-circ-CREBBP group. It has been reported that MCL1 and BCL2 inhibit apoptosis by binding with BAX[63,64]. Thus, we detected the amount of BAX binding to MCL1 and BCL2 in each group. The results showed that BAX expression significantly decreased in the Si-circ-CREBBP group. Consistent with these findings, the expression level of CASP3 was significantly increased in sperm with Si-circ-CREBBP, which promoted sperm apoptosis (Fig. 8). Moreover, the immunofluorescence results demonstrated that both MCL1/BAX and CREBBP/CREB1 were mainly colocalized in the tail of sperm cells. Therefore, the changes in the expression of these genes may directly affect the sperm motility.

In conclusion, our study is the first time to characterize the circRNAs present in boar SPEVs. We provided a comprehensive view of SPEVs circRNAs related to sperm motility and demonstrated that circ-CREBBP significantly increased sperm motility. In addition, circ-CREBBP targeted miR-10384 and miR-143-3p and inhibited sperm apoptosis by enhancing the expression of CREB1, CREBBP, MCL1, and BCL2 and reducing the expression of the proapoptotic factors BAX and CASP3. The clarification of the function of circ-CREBBP helps us to further understand the molecular mechanisms by which EVs affect sperm motility. Our study suggests that circ-CREBBP may be a promising biomarker and therapeutic target for male reproductive diseases.

## Methods
**Ethics statements**. All protocols for the collection of semen samples were reviewed and approved by the Committees for Ethical Review of China Agricultural University.

**Animals**. Using a computer-assisted sperm analysis (CASA) system (IVOS II, France), the sperm motility phenotype of 230 large white boars from a national boar station were collected. Twelve large white boars with high sperm motility and twelve large white boars with low sperm motility were selected for SPEV extraction. All the individuals were sexually mature, aged between 14 and 36 months old (Supplementary Table 3).

**Sample collection and Isolation of SPEVs**. Specialized professionals obtained the sperm-rich fractions of the ejaculate sample from each boar by the gloved hand method, and these samples were immediately assessed for sperm motility on a CASA instrument (IVOS II, France). SPEVs were isolated by ultracentrifugation method (EV-TRACK ID: EV200103)[24]. Thirty-five milliliters of semen plasma for each sample were used for EV isolation. Briefly, the cellular debris and impurity were removed by centrifuging at $10,000 \times g$ for 30 min at 4 °C and $12,000 \times g$ for 60 min at 4 °C. SPEVs were isolated by twice ultracentrifugation at $12,0000 \times g$ and 4 °C (Beckman, USA) and purified by 0.22 μm filters (Millipore, USA).

**Characterization of SPEVs**. Follow previous study, the morphology of SPEVs was observed by Transmission Electron Microscopy (TEM) and the size of SPEVs was detected by Nanoparticle Tracking Analysis (NTA). Furthermore, western blotting analysis of Calnexin (10,427–2-AP, Promega, Madison, WI), a negative marker of EVs, and EV markers protein such as anti-Alix (sc-53540, Santa Cruz, CA, USA), anti-Tsg101 (14497-1-AP, Proteintech), anti-CD9 (20597-1-AP, Proteintech), anti-CD81 (66866-1-AP, Proteintech) was used.

**RNA extraction and deep RNA sequencing**. SPEVs total RNA were isolated by using a miRNAeasy Mini Kit (Qiagen, Germany) according to the instructions. RNA quality, concentration and integrity were verified through 1% agarose gel electrophoresis and Agilent biological analyzer 2100 system (Agilent Technologies, CA, USA). Whole transcriptome sequencing was performed to gain insight into the types of RNAs in SPEVs, including miRNAs, mRNAs and circRNAs. Small RNA and long RNA libraries were established. Long RNA libraries were generated using the SMARTer Stranded Total RNA-Seq Kit (Takara Bio Inc.) according to the manufacturer's instructions, and the index code was added to the attribute sequences of each sample. Small RNA sequencing libraries were generated using a QIAseq miRNA Library Kit (Qiagen, Frederick, MD) following the manufacturer's recommendations, and the index code was added to the attribute sequence of each sample. The quality of the sequencing library was evaluated on an Agilent Analyzer 2100 and qPCR. A Truseq PE cluster kit v3 CBOT HS (Illumina, San Diego, CA, USA) was used to cluster index coding samples on the acbot cluster generation system. Finally, the library of each sample was sequenced on an Illumina HiSeq platform to generate paired-end reads.

Sperm RNA was extracted using TRIzol™ Reagent (Invitrogen, United States). The sperm sample were lysed with TRIzol for 15 min at room temperature. The samples were added with chloroform and centrifuged at 4 °C for 15 min at 12,000 rpm. Then, the top layer of clear liquid was added to an equal volume of isopropanol and centrifuged. The precipitate was washed with 75% alcohol and acryl carrier and then centrifuged. RNase-free water was added to dissolve the precipitate prior to storage at −80 °C. The RNA quality, concentration and integrity were verified through 1% agarose gel electrophoresis and Nanodrop 2000.

**CircRNA detection and annotation**. CircRNAs were identified by find_circ[29] and CIRI2[65] software. The overlapping results of two software were considered candidate identified circRNAs. Candidate circRNAs that contained at least 2 backsplice junction reads were filtered. Compared the location of reference genome genes, detected circRNAs were divided into three categories: exonic, intronic and intergenic regions. The relative expression abundance of each circRNA was estimated by RPM: Normalized expression = (mapped reads)/(total reads) × 1,000,000[66].

**CircRNA conservation and sequence feature analysis**. In brief, sequences of back-spliced exons were extracted, and identified homologous locations between humans (hg38 genome) and pigs (Sscrofa11.1 genome) by LiftOver tool. In addition, a human- and pig-expressed circRNA identified in pig/human orthologous locus without nucleotide difference in back-splice junction (BSJ) contexts (±5 nucleotides) was suggested as a conserved circRNA[67]. Then, PhastScore software was used to score the homologous regions identified between humans and pigs, and a score greater than 0.3 was considered more conservative. For identification of reverse complementary matches (RCMs), we aligned two intronic sequences flanking the same exonic circRNA using Basic Local Alignment Search Tool (BLAST). The flank introns of an exonic circRNA were input as the query sequence and subject sequence, respectively. For each intron pair, several alignments were obtained, and the alignment with the lowest e-value and the highest bitscore was regarded as reverse complementary matches (RCMs). Then, bedtools software with 'intersect' was applied to compare the RCM genomic locations with those of pig repeat elements downloaded from UCSC (http://hgdownload.cse.ucsc.edu/goldenPath/susScr11/database/rmsk.txt.gz).

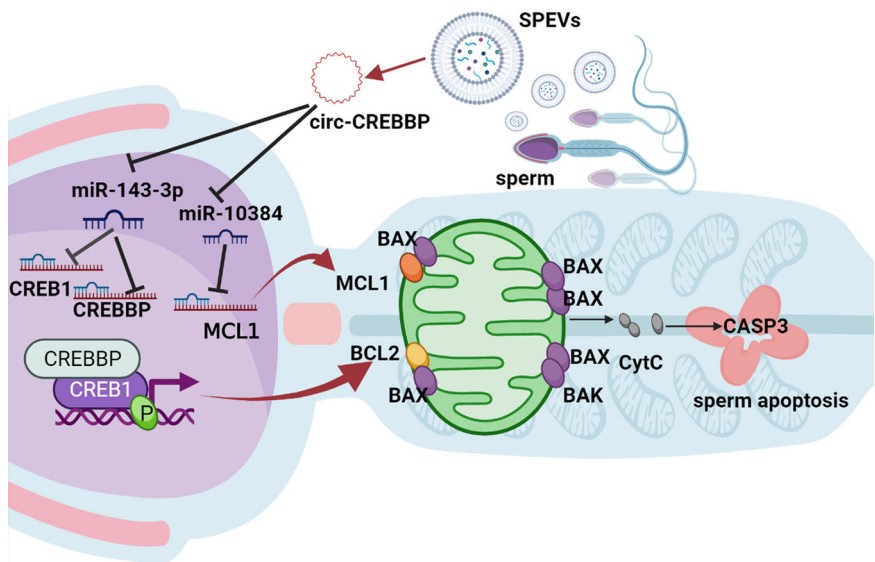

**Fig. 8 Schematic diagram of pathways by which circ-CREBBP regulates motility and apoptosis of porcine sperm.** The graphic was created with BioRender.com.

**Identification of miRNAs**. Details of miRNA identification are described in our previous study[25]. First, we used cutadapt software to trim adapter sequences of 3′ reads and then kept sequences with lengths between 18 and 32 nt after trimming[68]. To filter ncRNAs, such as rRNA, tRNA, snRNA and snoRNA, as well as repeated sequences, Bowtie was used to align clean data to the Silva[69], GtRNAdb[70], Rfam[71], and Repbase databases[72]. Then, the filtered sequences were further identified miR-NAs through miRDeep2[73] according to the following steps: (1) alignment to the pig reference genome (ftp://ftp.ensembl.org/pub/release-97/fasta/sus_scrofa) with no mismatch; (2) alignment to known mature and precursor miRNA sequences downloaded from the miRbase database (v21) (http://www.mirbase.org/).

**Differential expression analysis and functional enrichment analysis**. Differentially expressed circRNAs (DECs) between the high sperm motility group and the low sperm motility group were identified by the edgeR R package (v3.30.3) based on junction read counts. CircRNAs with a $P$ value < 0.05 and fold change (FC) ≥ 1.5 or ≤0.67 were considered DECs. We identified differentially expressed miRNAs (DEMs) through the same screening criteria using edgeR R package (v3.30.3). Gene ontology (GO) enrichment and KEGG pathway analysis for host genes of DECs and target genes of DEMs were established by the KOBAS website (http://kobas.cbi.pku.edu.cn/).

**ceRNA Network Construction**. Combined with the binding relationship between the identified DECs and DEMs, DEMs and target genes, a ceRNA network was constructed to reveal their potential interactions in porcine SPEVs. The DEM binding sites of DECs were predicted using miRanda software with criteria of score ≥ 150 and free energy ≤−7, and the results of RNAhybrid were combined with energy <−7 and the binding region in the seed region (2-8 nt) of miRNA. We further filtered the expression trends of DECs and DEMs and maintained the opposite expression trend of DECs and DEMs. Target genes of DEMs were predicted by miRanda and RNA-hybrid software with energy <−20. The ceRNA networks related to DECs were constructed and visualized using Cytoscape software v.3.5.0.

**Real-time quantitative PCR**. Total RNA was extracted from spermatozoa using TRIzol™ Reagent (Invitrogen, United States) according to the manufacturer's instructions. For qPCR of mRNA, 500 ng RNA was directly reverse transcribed using PrimeScript™ RT reagent Kit with gDNA Eraser (Takara Biotechnology, China). For qPCR of circRNA, 500 ng RNA was directly reverse transcribed using PrimeScript™ RT Master Mix (Takara Biotechnology, China) containing random and oligo (dT) primers. For qPCR of miRNA, 500 ng RNA was directly reverse transcribed using PrimeScript™ RT Master Mix (Takara Biotechnology, China) containing corresponding stem-loop primers. Quantitative PCR was then conducted on a CFX96 qPCR system (BioRad, USA) with TB® Green Master Mix (Takara Biotechnology, China) according to the manufacturer's instructions. The expression of miRNAs was standardized with U6, and the expression of mRNAs and circRNAs was standardized with glyceraldehyde phosphate dehydrogenase (*GAPDH*) and actin beta (*ACTB*), respectively. All primers used for qPCR are listed in Supplementary Table 4.

**Validation of the circular structure of circRNA**. Total RNA was extracted from SPEVs using a miRNAeasy Mini Kit (Qiagen, Germany). Then, RNase R treatment was carried out for 15 min at 37 °C using 0 U, 0.2 U, 0.4 U, and 1 U of RNase R

(Lucigen, US), respectively. The treated 500 ng RNA was directly reverse transcribed using PrimeScript™ RT Master Mix (Takara Biotechnology, China), and qPCRs were conducted using TB® Green Master Mix (Takara Biotechnology, China) according to the manufacturer's instructions.

**Luciferase reporter assay**. The wild-type gene of circ-CREBBP was amplified by PCR with a forward primer containing a SacI site and a reverse primer containing a XbaI site using PrimeSTAR polymerase (Takara Biotechnology, China) (Supplementary Table 5) and purified with a TIANgel Midi Purification Kit (Tiangen, China). The products were linked to the pmirGLO double luciferase reporter vector. The constructs were validated by sequencing. The mutation primers were designed to change the seed sequences of miRNA-gene binding sites to created Mutant vectors (Supplementary Table 5). The resulting reporter vectors are termed circ-CREBBP-WT, circ-CREBBP-MUT1 and circ-CREBBP-MUT2. According to the same method above, we constructed another six vectors: MCL1-WT and MCL1-MUT, CREB1-WT, CREB1-MUT, CREBBP-WT, and CREBBP-MUT. The recombinant plasmid was cotransfected with the miR-10384 mimic or inhibitor and miR-143-3p mimic or inhibitor into 293 T cells using Lipofectamine 2000 (Invitrogen, USA). After 48 hours of transfection, luciferase activity was detected using the Dual Luciferase Reporter Assay System (Promega, USA) by an enzyme labeling instrument (Tecan, Switzerland).

**SPEV electrotransfection and coincubation experiment**. To investigate the function of circ-CREBBP, miR-10384 and miR-143-3p in SPEVs on sperm motility, we coincubated sperm cells with electroporated SPEVs. SPEVs were electroporated with Si-circRNA, miRNA mimic or miRNA inhibitor using a Gene Pulser Xcell electroporator (Bio–Rad, USA) with the "exponential" protocol (250 V, 100 μF, R = ∞, cuvette size = 4 mm)[74]. The SPEVs were electroporated with the miR-10384 mimic (sense: 5′- CCCUG CGUGG CUUCU CUGUG CA -3′; antisense: 5′- CACAG AGAAG CCACG CAGGG UU -3′) or the miR-10384 inhibitor (sense: 5′-UGCAC AGAGA AGCCA CGCAG GG- 3′). The SPEVs were electroporated with the miR-143-3p mimic (sense: 5′-UGAGA UGAAG CACUG UAGCU C-3′; antisense: 5′-GCUAC AGUGC UUCAU CUCAU U-3′) or the miR-143-3p inhibitor (sense: 5′- GAGCU ACAGU GCUUC AUCUC A -3′). In addition, the SPEVs were electroporated with Si-circ-CREBBP (sense: 5′- GCGAA ACCAA CAAAU CUCAT T -3′; antisense: 5′- UGAGA UUUGU UGGUU UCGCT T -3′). The sperm were separated from semen samples by centrifuging at 800×$g$ for 10 min at 17 °C, and the concentration was adjusted to 1 × 10⁸ sperm/mL. Finally, we coincubated sperm cells with electrotransfected SPEVs at 17 °C. In coincubation experiments, sperm motility was determined visually using the CASA instrument (IVOS II) from day 1 to day 4. Sperm cells were collected on days 2 and 4 for qPCR and western blot verification.

**Western blot analysis and co-immunoprecipitation assay**. All SPEV samples and sperm samples were lysed with RIPA buffer (Solarbio, Beijing, China) containing 1% protease inhibitor, and the protein samples were quantified with a BCA assay kit (Beyotime, Beijing, China). The samples were separated by SDS–PAGE and then transferred to PVDF membranes (Millipore, United States). The PVDF membranes were blocked with 5% (w/v) skim milk, and incubated with antibodies against proteins of EVs markers and target genes, including anti-Alix (sc-53540,

Santa Cruz, CA, USA), anti-Tsg101 (14497-1-AP, Proteintech), anti-CD9 (20597-1-AP, Proteintech), anti-CD81 (66866-1-Ig, Proteintech), anti-Calnexin (10427-2-AP, Proteintech), anti-MCL1 (T55199, Abmart, Shanghai, China), anti-CREBBP (PA1273, Abmart, Shanghai, China), anti-CREB1 (MB0153S, Abmart, Shanghai, China), anti-pCREB1 (sc-81486, Santa Cruz), anti-BCL2 (12789-1-AP, Proteintech, USA), anti-BAX (60267-1-Ig; Proteintech, USA), anti-CASP3 (ab13847, Abcam, United Kingdom), anti-β-tubulin (10068-1-AP, Proteintech, USA), anti-α-tubulin (11224-1-AP, Proteintech, USA), and then detected by enhanced chemiluminescence (ECL) system.

A total of 200 μg protein lysates were precleared with protein A + G agarose (Beyotime, Beijing, China) at 4 °C for 1 h prior to immunoprecipitation with anti-MCL1 (T55,199, Abmart), anti-BCL2 (12789-1-AP, Proteintech, USA) and anti-CREBBP (sc-7300, Santa Cruz, CA, United States) overnight at 4 °C, with gentle rotation in a rotation mixer. The protein-antibody complexes were incubated with protein A/G-Sepharose beads for 4 hours at 4 °C with rotation and then centrifuged at 1000×g for 5 minutes at 4 °C and washed three times with RIPA buffer at 1000×g for 5 minutes. Subsequently, the bound proteins were separated via 60 μL 1X SDS–PAGE. The protein samples (25 μL) were separated by SDS–PAGE, transferred to PVDF membranes and incubated with anti-Ubiquitin (sc-27128, Santa Cruz, CA, United States), anti-BAX (60267-1-Ig, Proteintech), anti-phospho-Creb1 (S133) (T55043, Abmart, Shanghai, China) and anti-CREB1 (P16220, Abmart, Shanghai, China) antibodies. Then, the membranes were incubated with secondary antibodies and detected with an ECL system.

**Immunofluorescence staining and confocal microscopy**. The sperm incubated with SPEV were collected by centrifugation, and 4% paraformaldehyde was added and coated on anti-stripping glass. The sperm cells were fixed with 4% paraformaldehyde for 20 min at room temperature, washed with PBS three times for 5 min, permeabilized with 0.1% Triton X-100 for 20 min, and blocked with 3% BSA (ST023, Beyotime) for 30 min at room temperature. Subsequently, the cells were incubated with primary antibodies against MCL1 (Abmart, Shanghai, China), BAX (Proteintech, USA), CREBBP (Abmart, Shanghai, China), and CREB1 (Abmart, Shanghai, China) overnight at 4 °C. After the primary incubation, the cells were incubated with secondary antibodies at room temperature for 1 hour. The nuclei were stained with DAPI (C1002, Beyotime, China) at room temperature for 15 min, and fluorescence images were observed under a confocal microscope (Nikon, Japan) with a 100× oil-immersion objective lens.

**Evaluation of cell apoptosis**. The spermatozoa were washed and then resuspended in PBS for cell counting. The sperm apoptotic rate was evaluated according to the protocols provided by the Annexin V-FITC Apoptosis Detection Kit (Beyotime, China). Generally, $1 \times 10^5$ sperm cells were diluted within buffer, and added with Annexin V-FITC and PI. The cell mixture was cultured at room temperature for 20 min and then analyzed by a BD LSRFortessa flow cytometer. The gating strategies are showed in Supplementary Fig. 5.

**Detection of cellular ATP levels**. The ATP levels in spermatozoa were determined using a luciferase-based enhanced ATP assay kit (Beyotime, China) and BCA assay kit (Beyotime, China). Briefly, after incubation with EV and EV electroporated with circ-CREBBP siRNA, spermatozoa were washed with PBS and lysed immediately in 200 μL lysis buffer. Then, the lysate was collected and centrifuged at 12,000×g for 5 min at 4 °C. Then, 20 μL of supernatant was added to the well containing 100 μL ATP detection working dilution. The luminescence was detected by a multifunctional microplate reader (Biotek, USA). The protein concentration of each group was measured, which was used to calibrate the ATP levels in cells.

**Statistics and reproducibility**. All the experimental data were analyzed using Student's t test and ANOVA using Statistical Package for the Social Sciences (SPSS) software for Windows, release 21.0 (SPSS, Chicago, IL, United States). The false detection rate (FDR) was controlled for multiple comparisons. Each experiment was repeated at least three times. All results are presented as the mean ± standard error of mean (SEM). P values less than 0.05, 0.01 and 0.001 were considered statistically significant.

**Reporting summary**. Further information on research design is available in the Nature Portfolio Reporting Summary linked to this article.

## Data availability

Raw- and processed sequencing data is available through Gene-Expression omnibus (GEO) accession number GSE216966. The source data behind the graphs are available in Supplementary Data 5.

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

## Acknowledgements

We are grateful to the reviewers of this manuscript for their constructive suggestions. The authors are also indebted to the molecular quantitative genetics team at China Agricultural University for their expertize. This study was supported financially by the Key Technologies R&D Program of Guangdong Province (2022B0202090001), the National Key Research and Development Program of China (2019YFE0106800) and Anhui Academy of Agricultural Sciences Key Laboratory Project (No.2021YL023).

## Author contributions

L.J. conceived and designed the study. N.D. and M.H. performed the experiments. Y.Z., C.Z., and J.C. analyzed the data. J.L., C.W., and Q.Z. provided technical support. N.D. and Y.Z. wrote the paper, and L.J. revised the manuscript. All authors read and approved the final manuscript.

## Competing interests

The authors declare no competing interests.
