## [Peer Review File · Communications Biology]

Reviewers' comments:

Reviewer #1 (Remarks to the Author):

The study entitled "Circ-CREBBP 1 inhibits sperm apoptosis via the PI3K-Akt signaling 2 pathway by sponging miR-10384 and miR-143-3p" evaluated circRNAs of seminal plasma extracellular vesicles from boars with low and high sperm motility, in order to (i) identify expressed circRNAs and (ii) understand their activity on sperm function. It is an interesting, comprehensive study, and I just have a few questions and comments before this article can be suitable for publication, as described below:

1) The study hypothesis is still a little bit unclear for me. Were boars used as models to understand human physiology or the aim here was to evaluate circRNAs role on boars fertility? Was this study's objective to go through circ-CREBBP activity on sperm function from the start or to identify, in an untargeted approach, all expressed circRNAs in seminal plasma extracellular vesicles and, from the obtained results, this circRNA was selected for downstream analyses? If this is the case, it is not clear why the authors have chosen to specifically evaluate more deeply circ-CREBBP and circ-EP300, and, of these, to go further on the analysis of miRNAs involved with circ-CREBBP. These points are essential to understand this study and they are not clearly addressed throughout the manuscript.

2) Which values were considered to classify boars as low or high motility?

3) For Figure legends, do not use shortenings. Also, when there are shortenings inside the figure, this should be fully described in the figure legend (for example, L1 and H1 in Figure 2B).

4) Figure 1C, both graphs for H- and L-groups seem exactly the same. Is that correct?

5) Figure 1D, please specify both in the figure and in the figure legend which sample was analyzed in each of the 4 wells represented here. Were H- and L-groups analyzed and showed here?

6) Figure 2D, please better explain what this figure means. What is cumulative distribution and RPM? "The expression levels of circRNAs originating from exons, introns and intergenic regions did not show significant changes" – changes between groups?

7) What is the goal in evaluating the homology between pig and human circRNAs?

8) Figure 3A does not say much, and, in my opinion, it is unnecessary.

9) Figure 4, it is not clear the difference between top and bottom figures, because if a circRNA is upregulated in the L group, this means it is downregulated in the H group, and vice versa.

Reviewer #2 (Remarks to the Author):

This manuscript comprehensively profiled the expression of circRNAs and miRNAs in high and low large white boar sperm motility, and further explored the molecular mechanism of circ-CREBBP on sperm apoptosis via PI3K-AKT signaling pathways by sponging miR-10384 and -143-3p. This manuscript was well designed, written and provided enough data. However, the authors should revise their manuscript to address specific concerns before a final decision is reached.

Major concern:

In the introduction section, authors emphasized that they want to find some promising diagnostic biomarkers and therapeutic targets for male reproductive diseases. As we know, it exists an individual difference of sperm motility among different boar breeds, which was a great challenge for farmers to exclude those boars whose sperm quality or motility are quite low. These potential biomarkers with differentially expressed circRNAs or miRNAs in this manuscript were more prone to be used for

selection of boars with high sperm motility. The authors try to interpret these differentially expressed circRNAs or miRNAs to be a promising biomarker and therapeutic target for male reproductive diseases, but I can not totally agree with this point. From supplementary data 7, the mean sperm motility was about 40.09 ± 23.12 in the low sperm motility group. We cannot get more background information for the boars with low sperm motility, such as sperm motility in these boars was always low? Why these boars with low sperm quality were not excluded? Please confirm these boars with low sperm quality come from physiological or pathological ones.

Minor issue:

In Figure 5e and 5h, higher sperm motility was observed in NC and si-circ-CREBBP groups at 12h-96h. But lower survive rate of sperm were found in NC and si-circ-CREBBP groups at day2. Why?

Reviewer #1:

The study entitled “Circ-CREBBP1 inhibits sperm apoptosis via the PI3K-Akt signaling pathway by sponging miR-10384 and miR-143-3p” evaluated circRNAs of seminal plasma extracellular vesicles from boars with low and high sperm motility, in order to (i) identify expressed circRNAs and (ii) understand their activity on sperm function. It is an interesting, comprehensive study, and I just have a few questions and comments before this article can be suitable for publication, as described below:

1) The study hypothesis is still a little bit unclear for me. Were boars used as models to understand human physiology or the aim here was to evaluate circRNAs role on boars fertility? Was this study's objective to go through circ-CREBBP activity on sperm function from the start or to identify, in an untargeted approach, all expressed circRNAs in seminal plasma extracellular vesicles and, from the obtained results, this circRNA was selected for downstream analyses? If this is the case, it is not clear why the authors have chosen to specifically evaluate more deeply circ-CREBBP and circ-EP300, and, of these, to go further on the analysis of miRNAs involved with circ-CREBBP. These points are essential to understand this study and they are not clearly addressed throughout the manuscript.

Response: Thanks for your insightful comments. Many studies showed that pigs can be an ideal model for human biomedical research (Luo et al., 2012; Nagashima et al., 2012; Swindle et al., 2012). Therefore, studying complex human disease using pig models is a good strategy. In human, some male reproductive diseases, such as asthenospermian, oligospermia and prostate cancer associated with poor sperm motility and function. In this study, all expressed circRNAs of SPEVs of boars with high or low sperm motility were identified to screen important circRNAs affecting sperm motility. What attracted our attention was that the differential expression fold change of Circ-CREBBP in the two groups was in the top 5. In addition, based on the results of DEC targeting miRNAs and the opposite expression trend of circRNAs and miRNAs, we established the circRNA–miRNA–mRNA network. We found that

circ-CREBBP and circ-EP300 target miR-10384, miR-143-3p and miR-96-5p, respectively. Interestingly, these miRNAs are tightly linked by *EP300*, *CREB1*, *CREBBP*, *TLR4*, etc. We studied the homology between these circRNAs and miRNAs in human and pig, and found that they had high homology. The sequence homology of circ-CREBBP between pig and human is 90.1%. Previous studies showed CREBBP is highly expressed in spermatogonia and early spermatocytes (Boussouar et al., 2014). Moreover, CREBBP/P300 regulates a specific metabolic state both in progenitor spermatogenic cells and in late transcriptionally active spermatids (Boussouar et al., 2014). CREBBP and EP300 also play an important role in prostate cancer (Furlan et al., 2021). These data indicated that circ-CREBBP may affect sperm motility, so we chose it for further functional verification. We believe that the findings of this study have implications for human sperm reproduction. Please refer to lines 95-106 on page 4.

Reference:

- Boussouar, F., A. Goudarzi, T. Buchou, H. Shiota, S. Barral, A. Debernardi, P. Guardiola, P. Brindle, G. Martinez, C. Arnoult, S. Khochbin, and S. Rousseaux. 2014. A specific CBP/p300-dependent gene expression programme drives the metabolic remodelling in late stages of spermatogenesis. *Andrology*. 2:351–359. doi:10.1111/j.2047-2927.2014.00184.x.
- Furlan, T., A. Kirchmair, N. Sampson, M. Pühr, M. Gruber, Z. Trajanoski, F.R. Santer, W. Parson, F. Handle, and Z. Culig. 2021. MYC-Mediated Ribosomal Gene Expression Sensitizes Enzalutamide-resistant Prostate Cancer Cells to EP300/CREBBP Inhibitors. *Am J Pathol*. 191:1094–1107. doi:10.1016/j.ajpath.2021.02.017.
- Luo, Y., L. Lin, L. Bolund, T.G. Jensen, and C.B. Sørensen. 2012. Genetically modified pigs for biomedical research. *J Inherit Metab Dis*. 35:695–713. doi:10.1007/s10545-012-9475-0.
- Nagashima, H., H. Matsunari, K. Nakano, M. Watanabe, K. Umeyama, and M. Nagaya. 2012. Advancing pig cloning technologies towards application in regenerative medicine. *Reprod Domest Anim*. 47 Suppl 4:120–126. doi:10.1111/j.1439-0531.2012.02065.x.
- Swindle, M.M., A. Makin, A.J. Herron, F.J. Clubb, and K.S. Frazier. 2012. Swine as Models in Biomedical Research and Toxicology Testing. *Vet Pathol*. 49:344–356. doi:10.1177/0300985811402846.

2) Which values were considered to classify boars as low or high motility?

Response: In this study, the total sperm motility was measured using a computer-assisted sperm analysis (CASA) system (IVOS II, France), which is widely used in boar stations for sperm quality assessment. At present, the CASA system has been developed as an effective tool for rapid and objective assessment of sperm concentration, motility, kinematics and morphology in almost all mammals (van der Horst et al., 2018). According to the total sperm motilities of boars, the 24 individuals were divided into two groups. The boars in the H group had a higher total sperm motility (> 0.85), whereas the boars in the L group had a lower total sperm motility (< 0.73). The average total sperm motility of group H was 0.94, while that of group L was 0.40. Please refer to lines 115-116 on page 5.

Reference:

van der Horst G, Maree L, du Plessis SS. 2018. Current perspectives of CASA applications in diverse mammalian spermatozoa. *Reprod Fertil Dev* **30**:875–888. doi:10.1071/RD17468

3) For Figure legends, do not use shortenings. Also, when there are shortenings inside the figure, this should be fully described in the figure legend (for example, L1 and H1 in Figure 2B).

Response: Per your guidance, we have added the relevant description in the figure legends. Please refer to lines 882-883 and lines 887-888 on page 30.

4) Figure 1C, both graphs for H- and L-groups seem exactly the same. Is that correct?

Response: Sorry, it was a mistake. We corrected the picture of group L. Thanks!

5) Figure 1D, please specify both in the figure and in the figure legend which sample was analyzed in each of the 4 wells represented here. Were H- and L-groups analyzed and showed here?

Response: We have marked H- and L-groups on the Figure 1D. Thank you.

6) Figure 2D, please better explain what this figure means. What is cumulative

distribution and RPM? “The expression levels of circRNAs originating from exons, introns and intergenic regions did not show significant changes” – changes between groups?

Response: The circRNAs profiles were detected and the expression levels were calculated by RPM (spliced reads per-million) algorithm based on CIRI2 and find_circ software. The counts of the reads that spanned over back-splice junction sites were normalized to reads per million mapped reads (RPM) (Zhang et al., 2014; Wang et al., 2020). The cumulative distribution function reflects the proportion of circRNA expression abundance. In Figure 2D, different colors represent different circRNA types. Therefore, Figure 2D shows the proportion of different types of circRNAs in different expression abundance ranges. We observed that the circRNAs originated from exon, intron and intergenic region showed no significant changes in expression abundance.

Reference:

Wang, L., Z. You, M. Wang, Y. Yuan, C. Liu, N. Yang, H. Zhang, and L. Lian. 2020. Genome-wide analysis of circular RNAs involved in Marek’s disease tumorigenesis in chickens. *RNA Biol.* 17:517–527. doi:10.1080/15476286.2020.1713538.

Zhang, X.-O., H.-B. Wang, Y. Zhang, X. Lu, L.-L. Chen, and L. Yang. 2014. Complementary sequence-mediated exon circularization. *Cell.* 159:134–147. doi:10.1016/j.cell.2014.09.001.

7) What is the goal in evaluating the homology between pig and human circRNAs?

Response: Genes, miRNAs, lncRNAs and circRNAs with high homology in different species may have similar functions. We compared the sequences of circRNAs in pigs and humans, and found that 5,045 circRNAs showed no nucleotide difference in back-splice junction (BSJ) contexts (± 5 nucleotides). In addition, 2195 out of them are highly conserved in these two species, indicating that these conserved circRNA may play an important role in sperm function. In our further study, we focused on the molecular mechanism of circ-CREBBP affecting sperm motility. We first compared the sequences of circ-CREBBP in pigs and humans. Our results showed that the

homology of circ-CREBBP sequence between pigs and humans is 90.1%. Circ-CREBBP contains complementary binding sites with miR-143-3p and has a strong targeting relationship with miR-143-3p. Subsequently, we compared the sequences of miR-143-3p in a variety of species and found that the miR-143-3p is highly conserved in these species, including pigs and humans, indicating that circ-CREBBP may play a similar role in human sperm motility. Therefore, we think pigs are useful biomedical models for human biomedical research.

8) Figure 3A does not say much, and, in my opinion, it is unnecessary.

Response: We deleted it.

9) Figure 4, it is not clear the difference between top and bottom figures, because if a circRNA is upregulated in the L group, this means it is downregulated in the H group, and vice versa.

Response: Yes, we corrected the sentence (lines 902-904 on page 30). The top figure represents circRNAs upregulated in H group and miRNAs downregulated in the H group. The bottom figure represents circRNAs upregulated in L group and miRNAs downregulated in the L group. Thanks!

Reviewer #2:

This manuscript comprehensively profiled the expression of circRNAs and miRNAs in high and low large white boar sperm motility, and further explored the molecular mechanism of circ-CREBBP on sperm apoptosis via PI3K-AKT signaling pathways by sponging miR-10384 and -143-3p. This manuscript was well designed, written and provided enough data. However, the authors should revise their manuscript to address specific concerns before a final decision is reached.

Major concern:

In the introduction section, authors emphasized that they want to find some promising diagnostic biomarkers and therapeutic targets for male reproductive diseases. As we know, it exists an individual difference of sperm motility among

different boar breeds, which was a great challenge for farmers to exclude those boars whose sperm quality or motility are quite low. These potential biomarkers with differentially expressed circRNAs or miRNAs in this manuscript were more prone to be used for selection of boars with high sperm motility. The authors try to interpret these differentially expressed circRNAs or miRNAs to be a promising biomarker and therapeutic target for male reproductive diseases, but I can not totally agree with this point. Form supplementary data 7, the mean sperm motility was about 40.09 ± 23.12 in the low sperm motility group. We cannot get more background information for the boars with low sperm motility, such as sperm motility in these boars was always low?

Why these boars with low sperm quality were not excluded? Please confirm these boars with low sperm quality come from physiological or pathological ones.

Response: Thanks for your insightful comments. As you said, there are individual differences in sperm motility among different boar breeds. However, we assume that genes, molecules and pathways regulating the same biological functions or biological processes in the same species have common points and similarities. In fact, we conducted a similar study in Duroc boars and also found that the expression of circ-CREBBP in the high sperm group was higher than that in low sperm motility groups (see Figure 1 below, unpublished data). However, due to the small number of experimental individuals (only four individuals each group), it did not reach a significant level. These data suggested that there are some same regulatory molecules affecting sperm motility in different pig breeds. We believe that these common regulatory molecules are more important.

Figure 1 circ-CREBBP expression in Duroc
(H represents high sperm motility group, L represents low sperm motility group)

Regarding whether differentially expressed circRNAs or miRNAs can be used as potential biomarkers for male reproductive diseases, we have done in-depth analysis using miRNAs in our previous study (Zhang et al., 2021). We established a miRNA diagnosis model for prostate cancer (PCa) using 16 homologous differentially expressed miRNAs (DEmis). Our results showed that the combination of four DEmis exhibited an AUC of 0.914, suggesting that these DEmis can be used to distinguish PCa patients from controls. Most of these miRNAs are reportedly associated with PCa in humans. In addition, the role of circRNA as a miRNA sponge has been verified in many studies. In the current study, the high expression of circ-CREBBP is beneficial to maintain sperm viability and inhibits sperm apoptosis. On the contrary, when circ-CREBBP expression was down regulated, it was unfavorable to sperm viability. Some DECs (such as circ-SCAMP1 and circ-SLC22A3) identified in this study have been reported to be closely related to human PCa and can be used as diagnostic biomarkers (He et al., 2021; Wang et al., 2020). Therefore, these RNA molecules might be used as diagnostic biomarkers and potential therapeutic targets for male reproductive diseases.

Reference:

He, Y.-D., W. Tao, T. He, B.-Y. Wang, X.-M. Tang, L.-M. Zhang, Z.-Q. Wu, W.-M. Deng, L.-X. Zhang, C.-K. Shao, J. Zhou, L.-M. Rong, X. Gao, and L.-Y. Li.

2021. A urine extracellular vesicle circRNA classifier for detection of high-grade prostate cancer in patients with prostate-specific antigen 2–10 ng/mL at initial biopsy. *Mol Cancer*. 20:96. doi:10.1186/s12943-021-01388-6.

Wang, S., W. Su, C. Zhong, T. Yang, W. Chen, G. Chen, Z. Liu, K. Wu, W. Zhong, B. Li, X. Mao, and J. Lu. 2020. An Eight-CircRNA Assessment Model for Predicting Biochemical Recurrence in Prostate Cancer. *Front Cell Dev Biol*. 8:599494. doi:10.3389/fcell.2020.599494.

Zhang, Y., N. Ding, S. Xie, Y. Ding, M. Huang, X. Ding, and L. Jiang. 2021. Identification of important extracellular vesicle RNA molecules related to sperm motility and prostate cancer. *EVCNA*. 2:104–126. doi:10.20517/evcna.2021.02.

Regarding the selection of experimental individuals, all pigs were from a national boar station. Before semen collection and detection, it is impossible to distinguish between high sperm quality and low sperm quality, so potential individuals with low sperm quality cannot be excluded. In addition, these boars were examined by veterinarians and individuals without physiological or pathological diseases will be used to produce semen. In the 3 months before we collected the experimental samples, boars with high and stable sperm motility (total sperm motility ≥ 0.9) were selected as candidates for the high (H) sperm quality group. If the sperm motility of boars was about 0.6 for three consecutive tests, the boar station staff will pay special attention to these individuals. The sperm motility of these boars will continue to be tested for two months. Some of these individuals with persistently poor semen quality will be selected as candidates for the low (L) sperm motility group. As shown in Figure 2 below, the total sperm motility of 12 boars in L group was tested eight times within two months. The average total sperm motility of each individual was lower than 0.5. Because we have a cooperative relationship with the boar station, these individuals will be retained rather than eliminated.

Figure 2 The average total sperm motility of eight tests for each boar in L group within two months

Minor issue:

In Figure 5e and 5h, higher sperm motility was observed in NC and si-circ-CREBBP groups at 12h-96h. But lower survive rate of sperm were found in NC and si-circ-CREBBP groups at day2. Why?

Response: In our opinion, two different experimental methods may lead to significant changes in sperm motility. The sperm motility shown in Figure 5e were assessed using the CASA system (IVOS II, France) according to Springer Nature experiments (Mortimer and Mortimer, 2013). Sperm were incubated at 17 °C with diluent. After 2 mins of incubation at 37 °C, 7 µL of sperm suspension was placed on a prewarmed glass slide and covered with a glass cover slip. The detection will be completed within 30 s to 2 mins. There was no damage to sperm during the test. For Figure 5h, spermatozoa of NC and si-circ-CREBBP groups were washed with cold PBS, diluted in Annexin V-FITC binding buffer and stained with Annexin V-FITC and PI (C1062L, Beyotime, China). The sperm mixture was cultured at room temperature (25 °C) for 20 mins and then analyzed by a BD LSRFortessa flow cytometer. The detection process took about 60 mins. During the experiment, sperm were damaged by cold PBS, staining solution and centrifugation. Therefore, sperm motility in Figure 5h was significantly decreased.

Reference:

Mortimer D, Mortimer ST. 2013. Computer-Aided Sperm Analysis (CASA) of Sperm

Motility and Hyperactivation In: Carrell DT, Aston KI, editors.
Spermatogenesis: Methods and Protocols, Methods in Molecular Biology.
Totowa, NJ: Humana Press. pp. 77–87. doi:10.1007/978-1-62703-038-0_8

REVIEWERS' COMMENTS:

Reviewer #2 (Remarks to the Author):

All my previous concerns have been adequately addressed, and the article is now suitable for publication.

Reviewer #3 (Remarks to the Author):

The authors have addressed all comments carefully, I have no further comment and this manuscript can be accepted for publication.

Reviewer #1:

The study entitled “Circ-CREBBP1 inhibits sperm apoptosis via the PI3K-Akt signaling pathway by sponging miR-10384 and miR-143-3p” evaluated circRNAs of seminal plasma extracellular vesicles from boars with low and high sperm motility, in order to (i) identify expressed circRNAs and (ii) understand their activity on sperm function. It is an interesting, comprehensive study, and I just have a few questions and comments before this article can be suitable for publication, as described below:

1) The study hypothesis is still a little bit unclear for me. Were boars used as models to understand human physiology or the aim here was to evaluate circRNAs role on boars fertility? Was this study's objective to go through circ-CREBBP activity on sperm function from the start or to identify, in an untargeted approach, all expressed circRNAs in seminal plasma extracellular vesicles and, from the obtained results, this circRNA was selected for downstream analyses? If this is the case, it is not clear why the authors have chosen to specifically evaluate more deeply circ-CREBBP and circ-EP300, and, of these, to go further on the analysis of miRNAs involved with circ-CREBBP. These points are essential to understand this study and they are not clearly addressed throughout the manuscript.

Response: Thanks for your insightful comments. Many studies showed that pigs can be an ideal model for human biomedical research (Luo et al., 2012; Nagashima et al., 2012; Swindle et al., 2012). Therefore, studying complex human disease using pig models is a good strategy. In human, some male reproductive diseases, such as asthenospermian, oligospermia and prostate cancer associated with poor sperm motility and function. In this study, all expressed circRNAs of SPEVs of boars with high or low sperm motility were identified to screen important circRNAs affecting sperm motility. What attracted our attention was that the differential expression fold change of Circ-CREBBP in the two groups was in the top 5. In addition, based on the results of DEC targeting miRNAs and the opposite expression trend of circRNAs and miRNAs, we established the circRNA–miRNA–mRNA network. We found that

circ-CREBBP and circ-EP300 target miR-10384, miR-143-3p and miR-96-5p, respectively. Interestingly, these miRNAs are tightly linked by *EP300*, *CREB1*, *CREBBP*, *TLR4*, etc. We studied the homology between these circRNAs and miRNAs in human and pig, and found that they had high homology. The sequence homology of circ-CREBBP between pig and human is 90.1%. Previous studies showed CREBBP is highly expressed in spermatogonia and early spermatocytes (Boussouar et al., 2014). Moreover, CREBBP/P300 regulates a specific metabolic state both in progenitor spermatogenic cells and in late transcriptionally active spermatids (Boussouar et al., 2014). CREBBP and EP300 also play an important role in prostate cancer (Furlan et al., 2021). These data indicated that circ-CREBBP may affect sperm motility, so we chose it for further functional verification. We believe that the findings of this study have implications for human sperm reproduction. Please refer to lines 95-106 on page 4.

Reference:

- Boussouar, F., A. Goudarzi, T. Buchou, H. Shiota, S. Barral, A. Debernardi, P. Guardiola, P. Brindle, G. Martinez, C. Arnoult, S. Khochbin, and S. Rousseaux. 2014. A specific CBP/p300-dependent gene expression programme drives the metabolic remodelling in late stages of spermatogenesis. *Andrology*. 2:351–359. doi:10.1111/j.2047-2927.2014.00184.x.
- Furlan, T., A. Kirchmair, N. Sampson, M. Pühr, M. Gruber, Z. Trajanoski, F.R. Santer, W. Parson, F. Handle, and Z. Culig. 2021. MYC-Mediated Ribosomal Gene Expression Sensitizes Enzalutamide-resistant Prostate Cancer Cells to EP300/CREBBP Inhibitors. *Am J Pathol*. 191:1094–1107. doi:10.1016/j.ajpath.2021.02.017.
- Luo, Y., L. Lin, L. Bolund, T.G. Jensen, and C.B. Sørensen. 2012. Genetically modified pigs for biomedical research. *J Inherit Metab Dis*. 35:695–713. doi:10.1007/s10545-012-9475-0.
- Nagashima, H., H. Matsunari, K. Nakano, M. Watanabe, K. Umeyama, and M. Nagaya. 2012. Advancing pig cloning technologies towards application in regenerative medicine. *Reprod Domest Anim*. 47 Suppl 4:120–126. doi:10.1111/j.1439-0531.2012.02065.x.
- Swindle, M.M., A. Makin, A.J. Herron, F.J. Clubb, and K.S. Frazier. 2012. Swine as Models in Biomedical Research and Toxicology Testing. *Vet Pathol*. 49:344–356. doi:10.1177/0300985811402846.

2) Which values were considered to classify boars as low or high motility?

Response: In this study, the total sperm motility was measured using a computer-assisted sperm analysis (CASA) system (IVOS II, France), which is widely used in boar stations for sperm quality assessment. At present, the CASA system has been developed as an effective tool for rapid and objective assessment of sperm concentration, motility, kinematics and morphology in almost all mammals (van der Horst et al., 2018). According to the total sperm motilities of boars, the 24 individuals were divided into two groups. The boars in the H group had a higher total sperm motility (> 0.85), whereas the boars in the L group had a lower total sperm motility (< 0.73). The average total sperm motility of group H was 0.94, while that of group L was 0.40. Please refer to lines 115-116 on page 5.

Reference:

van der Horst G, Maree L, du Plessis SS. 2018. Current perspectives of CASA applications in diverse mammalian spermatozoa. *Reprod Fertil Dev* **30**:875–888. doi:10.1071/RD17468

3) For Figure legends, do not use shortenings. Also, when there are shortenings inside the figure, this should be fully described in the figure legend (for example, L1 and H1 in Figure 2B).

Response: Per your guidance, we have added the relevant description in the figure legends. Please refer to lines 882-883 and lines 887-888 on page 30.

4) Figure 1C, both graphs for H- and L-groups seem exactly the same. Is that correct?

Response: Sorry, it was a mistake. We corrected the picture of group L. Thanks!

5) Figure 1D, please specify both in the figure and in the figure legend which sample was analyzed in each of the 4 wells represented here. Were H- and L-groups analyzed and showed here?

Response: We have marked H- and L-groups on the Figure 1D. Thank you.

6) Figure 2D, please better explain what this figure means. What is cumulative

distribution and RPM? “The expression levels of circRNAs originating from exons, introns and intergenic regions did not show significant changes” – changes between groups?

Response: The circRNAs profiles were detected and the expression levels were calculated by RPM (spliced reads per-million) algorithm based on CIRI2 and find_circ software. The counts of the reads that spanned over back-splice junction sites were normalized to reads per million mapped reads (RPM) (Zhang et al., 2014; Wang et al., 2020a). The cumulative distribution function reflects the proportion of circRNA expression abundance. In Figure 2D, different colors represent different circRNA types. Therefore, Figure 2D shows the proportion of different types of circRNAs in different expression abundance ranges. We observed that the circRNAs originated from exon, intron and intergenic region showed no significant changes in expression abundance.

Reference:

Wang, L., Z. You, M. Wang, Y. Yuan, C. Liu, N. Yang, H. Zhang, and L. Lian. 2020. Genome-wide analysis of circular RNAs involved in Marek’s disease tumorigenesis in chickens. *RNA Biol.* 17:517–527. doi:10.1080/15476286.2020.1713538.

Zhang, X.-O., H.-B. Wang, Y. Zhang, X. Lu, L.-L. Chen, and L. Yang. 2014. Complementary sequence-mediated exon circularization. *Cell.* 159:134–147. doi:10.1016/j.cell.2014.09.001.

7) What is the goal in evaluating the homology between pig and human circRNAs?

Response: Genes, miRNAs, lncRNAs and circRNAs with high homology in different species may have similar functions. We compared the sequences of circRNAs in pigs and humans, and found that 5,045 circRNAs showed no nucleotide difference in back-splice junction (BSJ) contexts (± 5 nucleotides). In addition, 2195 out of them are highly conserved in these two species, indicating that these conserved circRNA may play an important role in sperm function. In our further study, we focused on the molecular mechanism of circ-CREBBP affecting sperm motility. We first compared the sequences of circ-CREBBP in pigs and humans. Our results showed that the

homology of circ-CREBBP sequence between pigs and humans is 90.1%. Circ-CREBBP contains complementary binding sites with miR-143-3p and has a strong targeting relationship with miR-143-3p. Subsequently, we compared the sequences of miR-143-3p in a variety of species and found that the miR-143-3p is highly conserved in these species, including pigs and humans, indicating that circ-CREBBP may play a similar role in human sperm motility. Therefore, we think pigs are useful biomedical models for human biomedical research.

8) Figure 3A does not say much, and, in my opinion, it is unnecessary.

Response: We deleted it.

9) Figure 4, it is not clear the difference between top and bottom figures, because if a circRNA is upregulated in the L group, this means it is downregulated in the H group, and vice versa.

Response: Yes, we corrected the sentence (lines 902-904 on page 30). The top figure represents circRNAs upregulated in H group and miRNAs downregulated in the H group. The bottom figure represents circRNAs upregulated in L group and miRNAs downregulated in the L group. Thanks!

Reviewer #2:

This manuscript comprehensively profiled the expression of circRNAs and miRNAs in high and low large white boar sperm motility, and further explored the molecular mechanism of circ-CREBBP on sperm apoptosis via PI3K-AKT signaling pathways by sponging miR-10384 and -143-3p. This manuscript was well designed, written and provided enough data. However, the authors should revise their manuscript to address specific concerns before a final decision is reached.

Major concern:

In the introduction section, authors emphasized that they want to find some promising diagnostic biomarkers and therapeutic targets for male reproductive diseases. As we know, it exists an individual difference of sperm motility among

different boar breeds, which was a great challenge for farmers to exclude those boars whose sperm quality or motility are quite low. These potential biomarkers with differentially expressed circRNAs or miRNAs in this manuscript were more prone to be used for selection of boars with high sperm motility. The authors try to interpret these differentially expressed circRNAs or miRNAs to be a promising biomarker and therapeutic target for male reproductive diseases, but I can not totally agree with this point. Form supplementary data 7, the mean sperm motility was about 40.09 ± 23.12 in the low sperm motility group. We cannot get more background information for the boars with low sperm motility, such as sperm motility in these boars was always low?

Why these boars with low sperm quality were not excluded? Please confirm these boars with low sperm quality come from physiological or pathological ones.

Response: Thanks for your insightful comments. As you said, there are individual differences in sperm motility among different boar breeds. However, we assume that genes, molecules and pathways regulating the same biological functions or biological processes in the same species have common points and similarities. In fact, we conducted a similar study in Duroc boars and also found that the expression of circ-CREBBP in the high sperm group was higher than that in low sperm motility groups (see Figure 1 below, unpublished data). However, due to the small number of experimental individuals (only four individuals each group), it did not reach a significant level. These data suggested that there are some same regulatory molecules affecting sperm motility in different pig breeds. We believe that these common regulatory molecules are more important.

Figure 1 circ-CREBBP expression in Duroc
(H represents high sperm motility group, L represents low sperm motility group)

Regarding whether differentially expressed circRNAs or miRNAs can be used as potential biomarkers for male reproductive diseases, we have done in-depth analysis using miRNAs in our previous study (Zhang et al., 2021). We established a miRNA diagnosis model for prostate cancer (PCa) using 16 homologous differentially expressed miRNAs (DEmis). Our results showed that the combination of four DEmis exhibited an AUC of 0.914, suggesting that these DEmis can be used to distinguish PCa patients from controls. Most of these miRNAs are reportedly associated with PCa in humans. In addition, the role of circRNA as a miRNA sponge has been verified in many studies. In the current study, the high expression of circ-CREBBP is beneficial to maintain sperm viability and inhibits sperm apoptosis. On the contrary, when circ-CREBBP expression was down regulated, it was unfavorable to sperm viability. Some DECs (such as circ-SCAMP1 and circ-SLC22A3) identified in this study have been reported to be closely related to human PCa and can be used as diagnostic biomarkers (He et al., 2021; Wang et al., 2020b). Therefore, these RNA molecules might be used as diagnostic biomarkers and potential therapeutic targets for male reproductive diseases.

Reference:

He, Y.-D., W. Tao, T. He, B.-Y. Wang, X.-M. Tang, L.-M. Zhang, Z.-Q. Wu, W.-M. Deng, L.-X. Zhang, C.-K. Shao, J. Zhou, L.-M. Rong, X. Gao, and L.-Y. Li.

2021. A urine extracellular vesicle circRNA classifier for detection of high-grade prostate cancer in patients with prostate-specific antigen 2–10 ng/mL at initial biopsy. *Mol Cancer*. 20:96. doi:10.1186/s12943-021-01388-6.

Wang, S., W. Su, C. Zhong, T. Yang, W. Chen, G. Chen, Z. Liu, K. Wu, W. Zhong, B. Li, X. Mao, and J. Lu. 2020b. An Eight-CircRNA Assessment Model for Predicting Biochemical Recurrence in Prostate Cancer. *Front Cell Dev Biol*. 8:599494. doi:10.3389/fcell.2020.599494.

Zhang, Y., N. Ding, S. Xie, Y. Ding, M. Huang, X. Ding, and L. Jiang. 2021. Identification of important extracellular vesicle RNA molecules related to sperm motility and prostate cancer. *EVCNA*. 2:104–126. doi:10.20517/evcna.2021.02.

Regarding the selection of experimental individuals, all pigs were from a national boar station. Before semen collection and detection, it is impossible to distinguish between high sperm quality and low sperm quality, so potential individuals with low sperm quality cannot be excluded. In addition, these boars were examined by veterinarians and individuals without physiological or pathological diseases will be used to produce semen. In the 3 months before we collected the experimental samples, boars with high and stable sperm motility (total sperm motility ≥ 0.9) were selected as candidates for the high (H) sperm quality group. If the sperm motility of boars was about 0.6 for three consecutive tests, the boar station staff will pay special attention to these individuals. The sperm motility of these boars will continue to be tested for two months. Some of these individuals with persistently poor semen quality will be selected as candidates for the low (L) sperm motility group. As shown in Figure 2 below, the total sperm motility of 12 boars in L group was tested eight times within two months. The average total sperm motility of each individual was lower than 0.5. Because we have a cooperative relationship with the boar station, these individuals will be retained rather than eliminated.

Figure 2 The average total sperm motility of eight tests for each boar in L group within two months

Minor issue:

In Figure 5e and 5h, higher sperm motility was observed in NC and si-circ-CREBBP groups at 12h-96h. But lower survive rate of sperm were found in NC and si-circ-CREBBP groups at day2. Why?

Response: In our opinion, two different experimental methods may lead to significant changes in sperm motility. The sperm motility shown in Figure 5e were assessed using the CASA system (IVOS II, France) according to Springer Nature experiments (Mortimer and Mortimer, 2013). Sperm were incubated at 17 °C with diluent. After 2 mins of incubation at 37 °C, 7 µL of sperm suspension was placed on a prewarmed glass slide and covered with a glass cover slip. The detection will be completed within 30 s to 2 mins. There was no damage to sperm during the test. For Figure 5h, spermatozoa of NC and si-circ-CREBBP groups were washed with cold PBS, diluted in Annexin V-FITC binding buffer and stained with Annexin V-FITC and PI (C1062L, Beyotime, China). The sperm mixture was cultured at room temperature (25 °C) for 20 mins and then analyzed by a BD LSRFortessa flow cytometer. The detection process took about 60 mins. During the experiment, sperm were damaged by cold PBS, staining solution and centrifugation. Therefore, sperm motility in Figure 5h was significantly decreased.

Reference:

Mortimer D, Mortimer ST. 2013. Computer-Aided Sperm Analysis (CASA) of Sperm

Motility and Hyperactivation In: Carrell DT, Aston KI, editors.
Spermatogenesis: Methods and Protocols, Methods in Molecular Biology.
Totowa, NJ: Humana Press. pp. 77–87. doi:10.1007/978-1-62703-038-0_8